# STEINSGATE: ADDING CAUSALITY TO DIFFUSIONS FOR LONG VIDEO GENERATION VIA PATH INTEGRAL

**Yufei Huang**[1,2], **Liangyu Yuan**[3], **Changxi Chi**[1,2], **Yunfan Liu**[1,2], **Cheng Tan**[4], **Siyuan Li**[1,2],
**Jingbo Zhou**[1,2], **Haitao Lin**[1,2], **Chang Yu**[2*], **Stan Z. Li**[2*]
[1] Zhejiang University, Hangzhou
[2] AI Lab, Research Center for Industries of the Future, Westlake University
[3] Shanghai JiaoTong University, Shanghai       [4]Shanghai Artificial Intelligence Laboratory
huangyufei@westlake.edu.cn; yuchang@westlake.edu.cn

## ABSTRACT

Video generation has advanced rapidly, but current models remain limited to short clips, far from the length and complexity of real-world narratives. Multi-action long video generation is thus both important and challenging. Existing approaches either attempt to extend the modeling length of video diffusion models directly or merge short clips via shared frames. However, due to the lack of temporal causality modeling for video data, they achieve only limited extensions, suffer from discontinuous or even contradictory actions, and fail to support flexible and fine-grained temporal control. Thus, we propose Instruct-Video-Continuation (*InstructVC*), combining Temporal Action Binding for fine-grained temporal control and Causal Video Continuation for natural long-term simulation. Temporal Action Binding decomposes complex long videos by temporal causality into scene descriptions and action sequences with predicted durations, while Causal Video Continuation autoregressively generates coherent video narratives from the text story. We further introduce SteinsGate, an inference-time instance of *InstructVC* that uses an MLLM for Temporal Action Binding and Video Path Integral to enforce causality between actions, converting a pre-trained TI2V diffusion model into an autoregressive video continuation model. Benchmark results demonstrate the advantages of SteinsGate and *InstructVC* in achieving accurate temporal control and generating natural, smooth multi-action long videos.

## 1 INTRODUCTION

Video is a central medium of modern culture, encompassing both professional productions (e.g., films, anime, television) and user-generated content (e.g., vlogs, fan-made animations). Video generation has thus emerged as a promising direction (Wan et al., 2025; Chen et al., 2025; Teng et al., 2025), aiming to lower creation barriers, expand narrative formats (e.g., interactive videos, memes), and improve creative efficiency (Bruce et al., 2024; HaCohen et al., 2024). The goal of video generation is to translate user-provided inputs—either textual narratives (Text-to-Video, T2V) or static frames (Image-to-Video, I2V)—into coherent visual stories. Despite recent advances that enable vivid short clips, current models are constrained to only a few seconds (Wan et al., 2025), far from the narrative length of real-world videos. This limitation motivates the study of long video generation, where models produce action-rich and narratively complete videos from a single prompt.

Long video generation faces two core challenges: long-term simulation, i.e., producing long, coherent, and **multi-action videos** beyond the current duration limits; and **temporal control**, i.e., accurately following action-rich prompts to ensure the correct order, completeness, and smoothness of actions. These challenges are inherently coupled: handling complex prompts often requires longer video sequences, while effective temporal control from prompts, in turn, reduces variance and guides the model to generate temporally consistent and well-connected action sequences.

---

*Corresponding Author

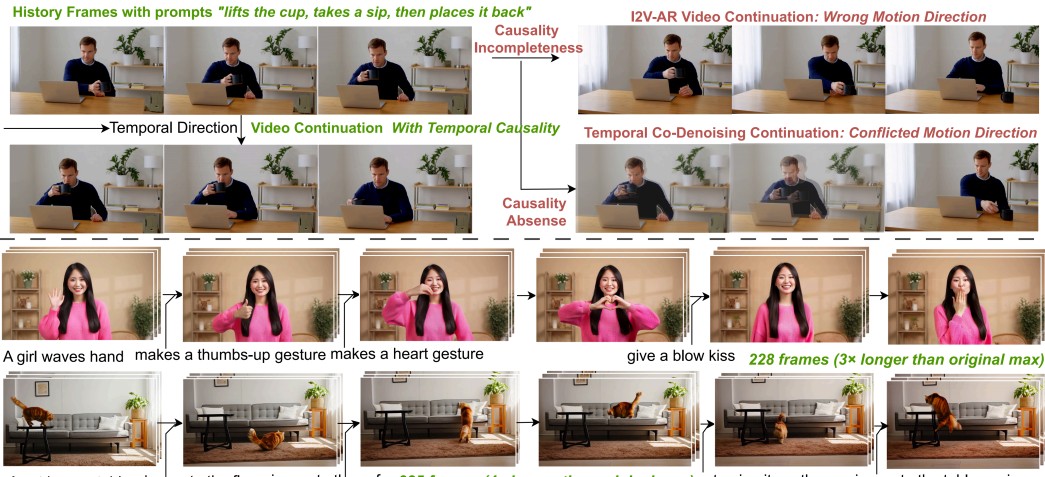

Figure 1: **Causal Video Continuation and Multi-action Long Video Generation.** *Upper panel*: Given instructions and history, our method captures temporal causality to continue videos smoothly, making multi-action transitions natural, like resuming a paused video. I2V-AR relies on the last frame and often misjudges motion direction, while Temporal Co-Denoising enforces correlation rather than causality via overlapping clips, causing conflicts in multi-action scenarios. *Lower panel*: SteinsGate achieves accurate action–time binding and smooth multi-action long video generation.

Existing methods for long video generation fall into two categories: temporal expanding (Lu et al., 2024; Kim et al., 2024) and temporal decomposition (Wang et al., 2023; Cai et al., 2025). Temporal expanding enlarges the token capacity of diffusion models (e.g. via frequency decomposition (Lu et al., 2024)), but can only extend length marginally. Thus, we follow the idea of temporal decomposition (TD), which breaks a long video into shorter segments, mitigating the length and control limits of temporal expanding. One framework for TD is temporal co-denoising, which generates each segment independently and enforces adjacent segment correlation (temporal correlation) through synchronized denoising of overlapping regions of adjacent segments (Lu et al., 2024; Cai et al., 2025). However, relying on temporal correlations without fully conditioning on previous segments (i.e., temporal causality) **often causes action direction conflicts** (Fig. 1). Since two adjacent segments generated independently, without respecting temporal causality, may exhibit completely mismatched action directions in their overlapping region. Another framework for TD is I2V-AR, which autoregressively generates each clip from only the last frame of the previous one. Only conditioning on last frame breaks temporal causality, as consecutive segments become only visually consistent while remaining blind to the dynamics of earlier segments. **This break often leading to motion reversal and poor temporal coherence** (Fig. 1). Finally, current TD methods only model local dependency between adjacent segments and ignore global causal planning, leading to incomplete sequences or broken causal order across multiple actions.

Motivated by the analysis above, we propose a new framework for multi-action long video generation, Instruct-Video-Continuation (*InstructVC*). It adds global and local causality in two stages (Fig. 3): Temporal Action Binding, focusing on causal temporal control to plan and place each action on a causal timeline, and Causal Video Continuation, focusing on temporal continuation to render the plans along the timeline. In Stage 1, given a user prompt, we enrich and decompose it into a scene and character description together with a sequence of action-duration pairs, to disentangle general motions into global causal action sequences. In Stage 2, a pretrained video diffusion model equipped with local temporal causality autoregressively continues the video based on current action descriptions and predicted action durations, completing each action before moving to the next if the last action duration is insufficient. Overall, *InstructVC* translates texts into videos in natural causal order. The firt stage acts like actors, *planning and performing actions along the timeline*, while the next stage *renders the video autoregressively*, producing the ongoing "performance".

We further introduce SteinsGate, a plug-and-play, inference-time instance of InstructVC that combines a Multi-modal Large Language Model (MLLM) for Temporal Action Binding and a novel temporal guidance technique, Video Path Integral, to enforce causality between action blocks and seamlessly convert a pre-trained TI2V diffusion model into an autoregressive video continuation

model. The Video Path Integral takes a short historical segment as input to enforce spatial and temporal causality. It samples historical frames as initial inputs for the TI2V model, predicts multiple future trajectories, and uses weighted integration to guide them toward the extended direction of the past. Leveraging the spatial–temporal disentanglement of I2V models, historical information is propagated into the continuation, making video generation process history-aware, temporally coherent, and autoregressively extendable while accurately following action sequences. To further improve the efficiency and effectiveness of Video Path Integral in practice, we introduce three optimizations in SteinsGate: (1) Guidance Interval, which reduces computation for path integral and improve efficiency; (2) History-aligned Redistribution, which promotes convergence along the extended direction of historical frames; and (3) Path Convergence Guidance, which strengthens the guidance progressively from weak to strong to better align generated video with historical context.

We leverage in-context learning on video dense-caption datasets (Wu et al., 2025), which provide real action sequences and durations, to teach an MLLM to enrich prompts and decompose them into detailed scenes with coherent action sequences and estimated durations. Binding actions to prompts with explicit durations—like holding a control key in a game for precise movement—reduces hallucinations and enables fine-grained temporal control, forming the basis for causal video continuation. During generation, the Video Path Integral prioritizes completing unfinished actions, ensuring smooth transitions to the next action after the current one is executed. To evaluate our framework, we construct the InstructVC Benchmark using generated multi-action storyboard-like prompts. Experiments show that SteinsGate and the InstructVC framework achieve accurate temporal control, smooth multi-action continuation, and natural long video generation, demonstrating our ability to translate textual narratives into coherent visual stories.

## 2 RELATED WORKS

**Video Generation with Diffusion Models** Research on video generation spans tasks, architectures, and generative frameworks. Text-to-video (T2V) generates videos from language (Chen et al., 2023; 2024b), while image-to-video (I2V) produces temporally coherent sequences from a single frame (Xing et al., 2023; Guo et al., 2023). I2V models are often considered spatial–temporal disentangled, injecting motion into the first frame and propagating its spatial information forward (Liu et al., 2025). Architecturally, early models relied on U-Net backbones (Chen et al., 2024b; Guo et al., 2023), but recent approaches have shifted to Diffusion Transformers (DiT) (Peebles & Xie, 2022; HaCohen et al., 2024; Wan et al., 2025). In terms of generative frameworks, autoregressive models suit streaming or interactive scenarios (Bruce et al., 2024; Chen et al., 2024a), while diffusion, especially DiT-based, dominates T2V and I2V (Yang et al., 2025; Team, 2024). Diffusion approaches often treat video as "3D images," ignoring its sequential and causal nature, which limits generalization to long or complex motions, hinders temporal control, and restricts video length.

**Long Video Generation** Many recent methods leverage pretrained diffusion models and decompose long video modeling with frequency or overlapping snippets (Cai et al., 2025; Wang et al., 2023). For example, FreeLong (Lu et al., 2024) uses spectral blending and local-global attention to combine low-frequency global structure with high-frequency local details without extra training, reducing high-frequency distortion. Gen-L-Video (Wang et al., 2023) processes overlapping short clips during denoising to produce long videos with diverse semantics while maintaining frame consistency. On the other hand, Autoregressive methods decompose long video into causally ordered short clips via the chain rule (Chen et al., 2024a; 2025; Teng et al., 2025), which aids control and modeling but suffers from error accumulation and is less compatible with non-causal pretrained video models (Kim et al., 2024). Inspired by these works, we propose to add causality into pretrained video diffusion foundation models at inference-time for plug-and-play temporal control and continuation. More related works could be found in the Appendix.

## 3 PRELIMINARIES

### 3.1 VIDEO GENERATION WITH DIFFUSION TRANSFORMERS

The *de facto* method to video generation is to encode videos into sequences of latent tokens and then apply diffusion modeling with transformer-based networks (Wan et al., 2025; HaCohen et al., 2024), commonly DiTs (Peebles & Xie, 2022). Given its scalability and strong performance, we adopt WanVideo 2.1 (Wan) (Wan et al., 2025), an open-source DiT-based model, as our pretrained foundation. Wan encodes an input video of frames $\mathbf{x} = \{x_i\}_{i=1}^{F}$ into latent tokens $\mathbf{z} = \{z_j\}_{j=1}^{N}$

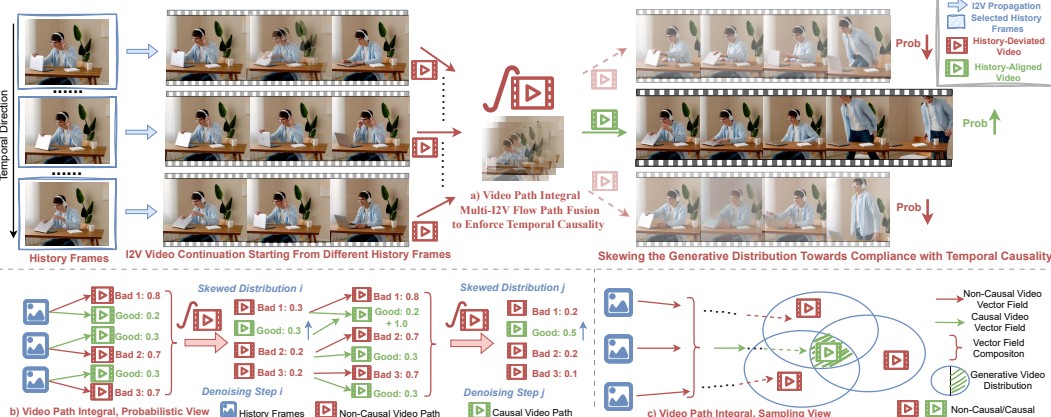

Figure 2: The illustration of Video Path Integral. a) We integrate over multiple I2V video paths (i.e., distributions or vector fields) of history frames to propagate not only spatial but also temporal information to continuing videos to add causality to pre-trained diffusion models b) During sampling, the probabilities of history-aligned videos reinforce each other due to their consistency, while history-deviated videos, being diverse, fail to reinforce and are gradually diluted. c) By conditioning on multiple historical frames, the continued video distribution is progressively constrained to satisfy historical conditions, approximating the true conditional distribution given the history.

using a 3D causal VAE with a spatiotemporal downsampling factor of 4×8×8. A denoising network $v_\theta$, implemented as an encoder-only transformer, then processes noisy latent tokens $\mathbf{z}_t$ together with text tokens $\mathbf{z}_{text}$ from text encoders via spatiotemporal self-attention for denoising and cross-attention for text alignment. The noisy latents are defined as $\mathbf{z}_t = (1-t)\epsilon + t\mathbf{z}$, where $\epsilon$ is a standard Gaussian noise and $t$ is the flow-matching timestep. Training follows the Flow Matching objective (Lipman et al., 2023), expressed as:

$$u(z_t, t|\epsilon, z_1) = u(z_t, t|z_1) = \frac{z_1 - z_t}{1 - t}, \quad \mathcal{L} = \mathbb{E}_{t, p_0(\epsilon), p_1(z_1)} \|v_\theta(z_t, t) - u(z_t, t|z_1)\|^2. \quad (1)$$

where $u(z_t, t|z_1)$ is the conditional velocity, representing a conditional video generation path. Then video generation takes the flow Ordinary Equation (ODE): $d\mathbf{z}_t = v(\mathbf{z}_t)dt, \mathbf{z}_0 \sim \mathcal{N}(\mathbf{0}, \mathbf{I})$.

## 3.2 MULTI-ACTION LONG VIDEO GENERATION

**Task Formulation**    Given a user-provided or extended prompt $P = [c_{txt}, c_{img}, \{a_i\}_{i=1}^N]$ containing the visual description $c_{txt}$, optionally an image condition $c_{img}$ as the first frame and ordered action descriptions $a_i$, i.e., textual narratives, our goal is to *translate* the text narrative into a video narrative by generating a long-term simulation that completes each action sequentially according to the temporal order in the prompt. Unlike mainstream video generation paradigms that follow image-generation formulations, we formalize long video generation as a translation task: akin to text translation, the target video is generated autoregressively by following the logical and sequential order of the source text, which explicitly requires temporal causality and continuity across actions.

**Compositional Generation**    Given a pretrained diffusion (or flow) sampler, one can sample from a single conditional distribution $p(x|c)$. But often we need to sample from the product of multiple conditionals—for example, conditioning jointly on both a pose $c_1$ and a reference image $c_2$-to get finer, more powerful control. Compositional Generation (Du et al., 2023) refers to methods that support sampling approximately from such product distributions. Its core idea is: given two pretrained distributions $p_\theta(x|c_1)$ and $p_\theta(x|c_2)$, with corresponding score functions $\nabla_x \log p(x|c_1)$ and $\nabla_x \log p(x|c_2)$, one can approximately sample from the product $p(x|c_1)p(x|c_2)$ by adding the score:

$$\nabla_x \log[p_\theta(x|c_1)p_\theta(x|c_2)] \approx \nabla_x \log p_\theta(x|c_1) + \nabla_x \log p_\theta(x|c_2). \quad (2)$$

The estimated score of the product distribution typically needs to be paired with more advanced samplers to enable more accurate sampling.

## 4 METHOD

### 4.1 INSTRUCT VIDEO CONTINUATION

**Motivations** Previous multi-prompt frameworks do not model temporal causality, instead using stitching techniques to merge independently generated clips. This creates a temporal coherence bottleneck, limiting flexible action transitions and making long videos appear as repeated edits of the same clip. Moreover, they neglect to construct prompts at the action level with distinct durations, typically assuming equal time spans for all prompts regardless of action complexity. This mismatch often causes actions to be skipped, incomplete, or repeated. In autoregressive generation, such errors accumulate, creating gaps between prompts (e.g., failing to walk to a table before being asked to pick up an item), which can ultimately collapse the generation.

We therefore propose the Instruct-Video-Continuation (InstructVC) framework (as shown in Fig. 3). Its core lies in Temporal Action Binding and Causal Video Continuation. Temporal Action Binding decomposes a long video into action-level units, predicts the duration of each, and binds them causally to the timeline. Guided by this plan, Causal Video Continuation autoregressively generates each action in sequence, enforcing temporal causality between actions.

**Temporal Action Binding** Given the strong text generation ability and rich world knowledge of MLLMs, we employ them as the executor of Temporal Action Binding. However, directly using an MLLM often introduces hallucinations: the decomposed action sequences may appear linguistically coherent but lack physical plausibility and diverge from the distribution of TI2V foundation model training data, where texts correspond to realistic videos. This mismatch leads to out-of-distribution prompts and poor video generation quality. To address this, we adopt in-context learning, providing examples from video dense caption datasets with multi-action prompts to guide the MLLM in leveraging its world knowledge for more realistic Temporal Action Binding. More details are provided in the Appendix.

**Causal Video Continuation** Guided by Temporal Action Binding, we explicitly model temporal causality through video continuation, generating each action sequentially in temporal order based on the history of previous ones. If

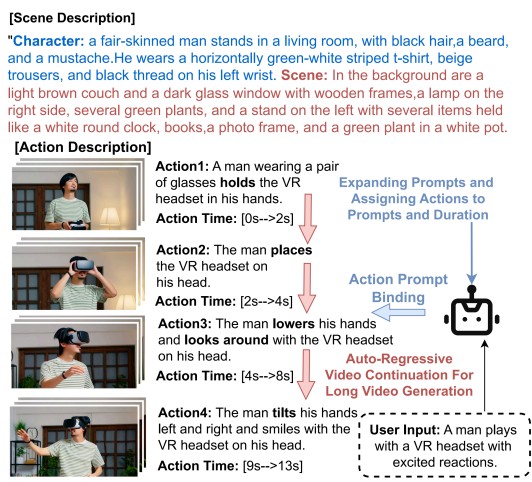

Figure 3: The *InstructVC* framework. An MLLM splits the user prompt into a scene description and coherent action sequences, refines each action to match model prompt style, and predicts its duration, producing a temporally grounded storyboard for next causal video continuation.

the previous action is unfinished, the continuation naturally completes it first—for example, closing a laptop left half-shut before standing up—thereby enforcing causal consistency. In the following, we describe how a pretrained video diffusion foundation model can be transformed into a Causal Video Continuation model in a plug-and-play manner at inference time.

### 4.2 VIDEO PATH INTEGRAL

To perform explicit temporal modeling for multi-action video, we aim to model the joint distribution of the video sequence. By the chain rule, this distribution can be factorized and simplified as:

$$p(\mathbf{z}_{1:N}) = \prod_{i=1}^{N} p(\mathbf{z}_i \mid \mathbf{z}_{<i}) \xrightarrow{\text{First-order Markov}} p(\mathbf{z}_{1:N}) \approx \prod_{i=1}^{N} p(\mathbf{z}_i \mid \mathbf{z}_{i-1}) \quad (3)$$

Here $a_{1:N}$ denotes the sequence of video segments (or action-level clips). The first-order Markov assumption approximates each segment as depending only on the immediately preceding one, which is a common simplification in autoregressive video generation to improve tractability while retaining temporal coherence (Bruce et al., 2024; Chen et al., 2024a). For simplicity, we will omit text or

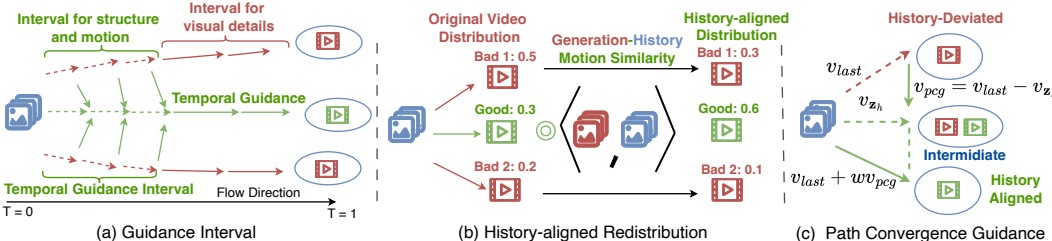

Figure 4: The illustration of SteinsGate framework. Our key designs are threefold: (a)Temporal guidance is applied only during the dynamics- and structure-controlling phase, improving sampling speed without sacrificing quality; (b) Path weights are adjusted based on the motion alignment between generated (the overlapped part) and real history, shifting the distribution toward history-consistent directions. (c)The sampling velocity is anchored to the last-frame I2V velocity to respect happened history, while a guidance difference gradually steers generation to add temporal causality.

image conditioning in the formulations of this section. That's said, given the pretrained video generation model $p_\theta(\mathbf{z}_i)$, we need to approximate the conditional distribution $p_\theta(\mathbf{z}_i \mid \mathbf{z}_{i-1})$. A common practical simplification is to assume that consecutive video segments share an overlapping history region $\mathbf{z}_h = \{\mathbf{z}_i \cap \mathbf{z}_{i-1}\}$, and the conditional distribution becomes $p_\theta(\mathbf{z}_i \mid \mathbf{z}_{i-1}) \approx p_\theta(\mathbf{z}_i \mid \mathbf{z_h})$.

**Limitations of Spatial-to-Temporal Guidance** Note that $\mathbf{z}_h$ is a subset of $\mathbf{z}_i$. A straightforward baseline is to cast enforcing temporal causality as a classical spatial inverse problem (Meng et al., 2022), which studies how to infer a complete sample given partial observations (e.g., the historical region) under consistency constraints. A well-studied solution is the Reconstruction Guidance technique (Chung et al., 2023), which gradually reconstructs the given portion of a sample over multiple sampling steps by introducing a reconstruction gradient. Under the flow matching framework, it can be formulated as (Pokle et al., 2023):

$$v_\theta(\mathbf{z}_t, t|\mathbf{z}_h) = v_\theta(\mathbf{z}_t, t) + \eta(t)\nabla_{\mathbf{z}_t} \log p(\mathbf{z}_h|\mathbf{z}_t), \quad \nabla_{\mathbf{z}_t} \log p(\mathbf{z}_h|\mathbf{z}_t) = \nabla_{\mathbf{z}_t}\|\mathbf{z}_h - \hat{\mathbf{z}}_h\|_2^2. \quad (4)$$

where $\hat{\mathbf{z}}_h$ is the predicted history given noisy $\mathbf{z}_t$ and $\eta(t)$ is a coefficient with respect to the timestep. Current video models treat videos as "3D images", making it reasonable to borrow spatial-domain techniques. However, even with advanced spatial guidance (as shown in Experiment Sec. 5.2), the generated samples can perfectly reconstruct the historical portion, yet the continued video exhibits noticeable gaps from history, showing that the temporal structure is not properly modeled. Despite representing video as 3D images, predicting future frames from history remains highly uncertain, and the success of local-to-global spatial guidance does not carry over to the temporal domain.

**Video Path Integral as Temporal Guidance** Observations above motivates our study of temporal guidance for history-to-future video generation, propagating historical information to influence the future. In addition to implicit methods like Reconstruction Guidance, we seek an explicit solution (Fig. 2). Given the TI2V models which propagate *spatial* information from the first frame via conditional vector fields, we define the resulting video distribution as the *I2V Video Path*. Our idea is to *integrate the I2V Video Paths* of historical frames—their I2V vector fields—during multi-step sampling of the continued video (as shown in Fig. 2, thereby explicitly propagating *spatio-temporal* information from history into continuation and extending I2V from Image-to-Video to History-to-Future at inference time. Video Path Integral could be formulated as:

$$v_\theta(\mathbf{z}_t, t|\mathbf{z}_h) = \int_{i=0}^{H} w_t(v_\theta)\hat{v}_\theta(\mathbf{z}_t, t|x_i)dx_i \approx \sum_{j=1}^{K} w_t(v_\theta)\hat{v}_\theta(\mathbf{z}_t, t|x_j), \ \{x\}_{j=1}^{K} \subset \{x\}_{i=1}^{H}. \quad (5)$$

where $\{x\}_{i=1}^{H}$ denotes the history images and $\{x\}_{j=1}^{K}$ the subset selected for *Monte-Carlo Estimation* due to frame rates and efficiency constrains in practice. And $w(v_\theta)$ is normalizing and temporal weighting factors for history alignment. $\hat{v}_\theta$ represents the velocity predicted after replacing the corresponding historical segments in the generated trajectory $\mathbf{z_t}$ with noisy real history $\mathbf{z}_t^h$, supplementing the image condition with dynamic and temporal information. For simplicity, we omit this notation in the subsequent analysis and more details are provided in the Appendix.

The core of Video Path Integral is how temporal information is propagated into the future. The key lies in the nested structure of time: the I2V Video Path starting from a history frame $x_j$ already includes the path from the subsequent frame $x_{j+1}$ and so on. When integrating across the I2V Video Paths of all history frames, the trajectories consistent with the entire history—i.e., those aligned

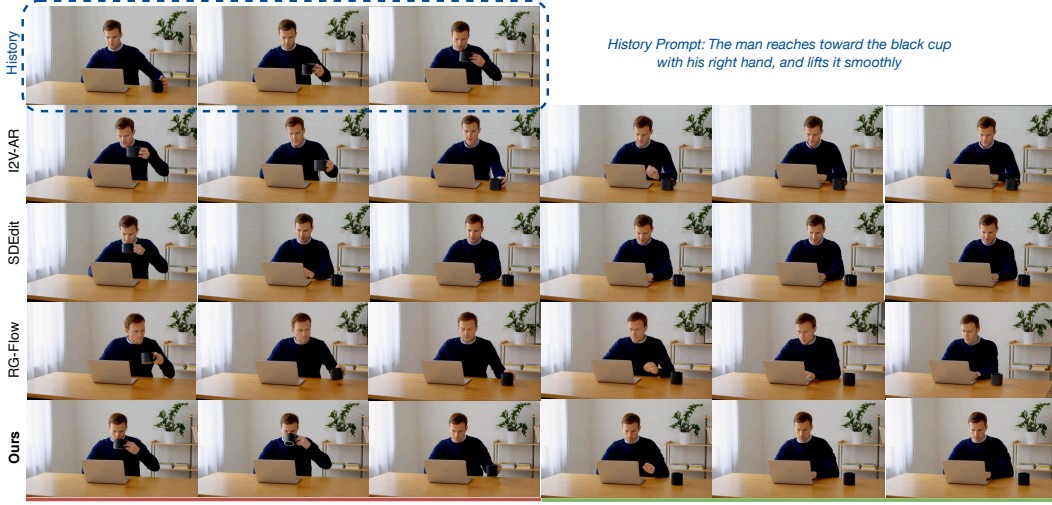

Figure 5: Qualitative Comparison for Video Continuation. Given prior history, SteinsGate follows the historical trajectory to complete the current action and transition smoothly to the next. Other causality-enforcing methods fail to propagate spatio-temporal information, often skipping required actions, reversing motion, or producing jumps between actions.

with temporal causality—are reinforced, while inconsistent ones, starting from different frames, are gradually diluted. This is analogous to the path integral in Quantum Physics, where path distributions strengthen along the correct macroscopic trajectory and cancel out along incorrect paths. As a result, the Video Path Integral converges toward the direction consistent with historical temporal causality. This can be further interpreted from both probabilistic and sampling perspectives:

$$\text{Probabilistic: } p(\mathbf{z}_i|\mathbf{z}_h) \propto \prod_{j=1}^{K} p(\mathbf{z}_i|x_j), \quad \text{Sampling:} \nabla_{\mathbf{z}_t} \log p(\mathbf{z}_i^t|\mathbf{z}_h) \approx \sum_{j=1}^{K} \nabla_{\mathbf{z}_t} \log p(\mathbf{z}_i^t|x_j). \quad (6)$$

That's, history conditional distribution is approximated by the product of frame-wise conditional distributions, enabling sampling via compositional generation (as Eq. 2). In the flow matching setting, the score is converted into a vector field and further details are given in the Appendix.

## 4.3 STEINSGATE

To improve sampling efficiency, enforce temporal coherence with historical context, and reduce the estimation error of compositional generation, we introduce three simple enhancements (as shown in Fig 4, resulting in a practical, plug-and-play causal video continuation method, SteinsGate.

**Guidance Interval.** Since Video Path Integral requires repeated velocity computations, we apply it only in the high-noise stage—where visual structure and motion are primarily determined—to improve sampling efficiency. In later stages, which mainly refine visual details without altering overall motion, we directly use the I2V vector field of the last historical frame.

**History-aligned Redistribution.** Except for the last historical frame, each I2V video path overlaps with the history to varying lengths. To encourage the generated video to converge along the history, we weight different I2V video paths based on the known history, biasing the intermediate video distribution toward alignment with it. To avoid interference from static regions and varying overlap lengths, we propose Motion-Aware History Shifting, which weights each path according to the dynamic similarity between its predicted historical trajectory and the ground-truth history:

$$w_t(v_\theta(\mathbf{z}_t, t \mid x_j)) = \text{cos-similarity}\langle \mathbf{m}_{j:H}^{v_\theta}, \mathbf{m}_{j:H}^{\mathbf{z}_H}\rangle, \quad \mathbf{m}_{j:H} = \mathbf{z}_{j+1:H} - \mathbf{z}_{j:H-1} \quad (7)$$

where $\mathbf{m}$ is the motion vector within the predicted history with $v_\theta$ and the true history.

**Path Convergence Guidance.** To reduce the estimation error of compositional generation, we adopt a more powerful sampling technique. Unlike traditional, time-consuming MCMC methods (Du et al., 2023), inspired by AutoGuidance (Karras et al., 2024), we propose Path Convergence Guidance(PCG): the I2V velocity of the last frame—without temporal causality—is used as the weak model estimate, while the result of Video Path Integral serves as the strong model estimate. Their

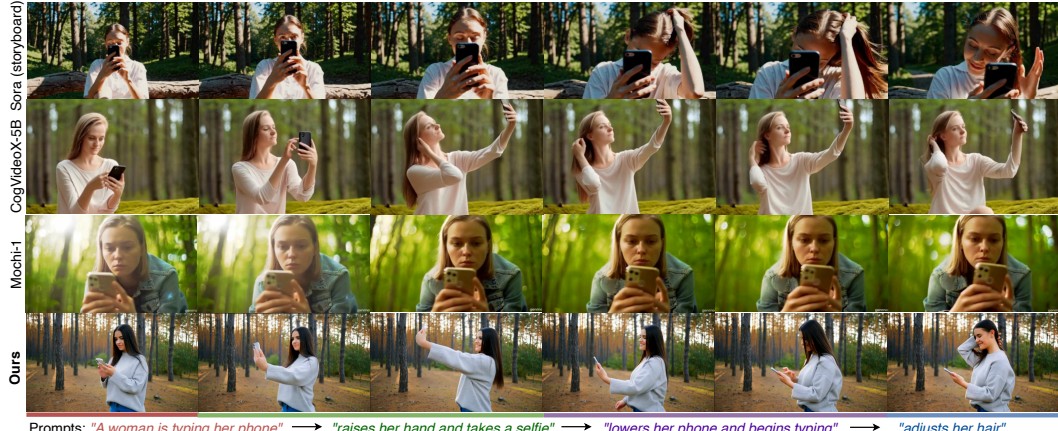

Prompts: *"A woman is typing her phone"* ⟶ *"raises her hand and takes a selfie"* ⟶ *"lowers her phone and begins typing"* ⟶ *"adjusts her hair"*

Figure 6: Multi-action Long Video Generation. We conduct a system-level comparison across diverse open-source and commercial models. Results show that SteinsGate achieves accurate action–time binding and supports coherent multi-action text-to-video narrative translation.

| | DiTCtrl | SkyReel-V2 | MAGI-1 | FIFO | **SteinsGate** | w/o VPI | w/o GI | w/o HR | w/o PCG |
|---|---|---|---|---|---|---|---|---|---|
| CSCV↑ | 0.76 | **0.83** | 0.82 | 0.71 | 0.82 | 0.74 | 0.81 | 0.79 | 0.78 |
| Motion Smoothness↑ | 0.93 | 0.96 | 0.96 | 0.89 | **0.97** | 0.94 | 0.97 | 0.95 | 0.96 |
| Text-Image Alignment↑ | 0.31 | **0.34** | 0.33 | 0.29 | 0.32 | 0.31 | 0.33 | 0.32 | 0.31 |

Table 1: Quantitative Comparison and Ablation Study. VPI denotes Video Path Integral, GI denotes Guidance Interval, and HR denotes History-aligned Redistribution. We compare SteinsGate with mainstream DiT-based autoregressive and Temporal Co-Denoising methods. SteinsGate outperforms other inference-time methods and achieves performance comparable to costly diffusion-forcing approaches. Each component contributes to our efficiency and effectiveness.

difference is then used as the weak-to-strong guidance velocity $v_{pcg} = v_\theta(\mathbf{z_t} \mid \mathbf{z}_h) - v_\theta(\mathbf{z}_t \mid x_{last})$. Combined with classifier-free guidance(CFG), our sampling procedure can be summarized as:

$$v_\theta^* = \begin{cases} v_\theta^{last} + w_1 v_{pcg} + w_2(v_\theta(\mathbf{z}_t \mid x_{last}) - v_\theta(\mathbf{z}_t|x_{last}, \emptyset) & \text{if } t \le t_{mid} \\ v_\theta^{last} + w_2(v_\theta(\mathbf{z}_t \mid x_{last}) - v_\theta(\mathbf{z}_t|x_{last}, \emptyset) & \text{if } t > t_{mid} \end{cases} \quad (8)$$

where $t_{mid}$ is the interval threshold (usually set to 0.3) and $v_\theta(\mathbf{z}_t \mid x_{last}, \emptyset)$ denotes the I2V velocity without text condition (for simplicity, we omit the text condition, assuming it is present unless specified), and $w_1, w_2$ are guidance strengths (usually set to 1.5 and 5.0 respectively) for PCG and CFG. Causal Video Continuation follows the ODE: $d\mathbf{z}_t = v_\theta^*(\mathbf{z_t}, \mathbf{z_h}, x_{1:K})dt, \ \mathbf{z}_0 = \epsilon \sim \mathcal{N}(\mathbf{0}, \mathbf{I})$.

## 5 Experiments

In this section, we conduct both qualitative and quantitative experiments across multiple tasks, including video continuation, action time binding, and multi-action long video generation, comparing against baselines from causality enforcing, temporal decomposition, and autoregressive approaches.

### 5.1 Experiments Setup

**Datasets.** We construct the InstructVC benchmark from video dense captions in MinT (Wu et al., 2025) and StoryBench (Bugliarello et al., 2023), which provide storyboard-like temporal captions with sequential actions. These prompts are further expanded and diversified to match the InstructVC format. To evaluate the capability of Temporal Action Binding and ensure broader coverage of diverse scenarios, we additionally expand short prompts from VBench (Huang et al., 2024).

**Baselines.** We primarily compare against autoregressive baselines that perform autoregression along the temporal axis while still generating frames with diffusion. For the video continuation, we additionally implement causality-enforcing baselines that constrain adjacent clips at inference time, including I2V-AR, an enhanced version of Reconstruction Guidance (Huang et al., 2025) under flow matching (RG-Flow), and the classic spatial-guidance method SDEdit (Meng et al., 2022). For temporal action binding and multi-action long video generation, we benchmark against diffusion-forcing–based text-to-video models (SkyReel-V2 (Chen et al., 2025), MAGI-1 (Teng et al., 2025))

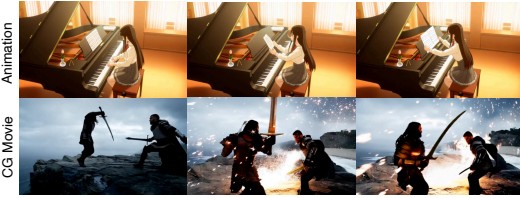

Prompts: *"A cat walks towards a bowl"* → *"drinks water"* → *"lifts its head"*   Top Panel: *"A girl plays piano"* Bottom Panel: *"Two warriors engage in a swordfight"*

Figure 7: Ablation Study and More Results. We concatenate action prompts as clip prompts and evenly distribute predicted durations for clip-by-clip autoregressive generation (w/o Action Binding).Results highlight the importance of Temporal Action Binding for precise temporal control and demonstrate the necessity and potential of MLLMs in handling more complex long video generation. SteinsGate also preserves the pretrained model's capabilities, supporting diverse video styles.

and the training-free FIFO-Diffusion (FIFO) (Kim et al., 2024). We also include temporal co-denoising methods, represented by DiT-based DiTCtrl (Cai et al., 2025), noting that many earlier approaches relied on U-Net backbones. Finally, we provide qualitative comparisons with additional open-source models (Mochi-1 (Team, 2024), CogVideoX-5B (Yang et al., 2025)) and the commercial Sora (Storyboard version). More details could be found in the Appendix.

## 5.2 VIDEO CONTINUATION

As a core task of the InstructVC framework, we conduct qualitative experiments on text-based video continuation. Using prompts from the InstructVC benchmark, we generate historical videos from the earlier action prompts and continue them with subsequent prompts. We implement causality-enforcing baselines on Wan2.1—the same as SteinsGate—by applying inference-time techniques to enforce causal continuity between the generated continuation and the history. Results (Fig. 5) show that SteinsGate successfully continues videos along the causal trajectory of the history and faithfully follows text instructions, while other methods often ignore required actions, produce motions opposite to the historical trend (the 2nd and 4th rows), or create discontinuities between history and continuation (the 3rd row).

## 5.3 MULTI-ACTION LONG VIDEO GENERATION

To evaluate multi-action long video generation, we perform a system-level comparison including both qualitative and quantitative experiments. Qualitatively, we focus on action time binding—executing each action within the specified duration. Results in Fig. 6 show that our method accurately generates the specified actions within the given time intervals, producing high-quality long videos with coherent and natural motions. In contrast, other methods struggle to generate actions precisely, often skipping actions or disrupting their temporal order. Quantitatively, we measure multi-action continuity and video quality following the DiTCtrl protocol in Tab. 1.. Metrics include the Clip Similarity Coefficient of Variation (CSCV) to assess transition smoothness, CLIP similarity to evaluate alignment between prompts and video clips, and VBench Motion Smoothness to assess whether generated motions are smooth and physically plausible. Results show that our inference-time method achieves performance comparable to costly training-based T2V diffusion-forcing methods, while outperforming other training-free approaches.

## 5.4 ABLATION STUDY

To illustrate the effect of Temporal Action Binding and the contributions of SteinsGate components, we compare against a global conditioning baseline (w/o Temporal Action Binding) where the scene description and multi-action prompts are concatenated and total action duration is evenly divided across segments for video continuation. Results in Fig. 7 show that Temporal Action Binding enables accurate temporal control, preventing skipped or misordered actions that cause discontinuities. Additional ablations in Tab. 1 confirm that removing Video Path Integral while keeping Temporal Action Binding with I2V-AR reduces temporal guidance effectiveness, whereas introducing the Guidance Interval preserves most performance while halving inference time. We also showcase additional videos in diverse styles (Fig. 7) to demonstrate SteinsGate's ability to preserve the capabilities of the pretrained model while supporting a wide range of user requirements.

## 6 CONCLUSION

We propose the InstructVC framework for multi-action long video generation, enabling stronger temporal control and more natural long-term simulation through Temporal Action Binding and Causal Video Continuation. We further introduce SteinsGate, an inference-time instance of InstructVC, which uses an MLLM and the temporal guidance technique Video Path Integral to inject causal awareness into pre-trained video diffusion models. A remaining limitation is long-term consistency, which is itself a highly challenging research direction, as our focus is on temporal causal continuity and thus maintaining coherence relies on selecting an appropriate history length.

## ACKNOWLEDGEMENTS

This work was supported by National Science and Technology Major Project (No.2022ZD0115101), National Natural Science Foundation of China Project (No.624B2115 No.U21A20427), Project (No. WU2022A009) from the Center of Synthetic Biology and Integrated Bioengineering of Westlake University and the Hangzhou Postdoctoral Daily Funding Program (No.103140026582502, 2025).

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

# A  APPENDIX

## A.1  MORE RELATED WORKS AND BACKGROUNDS

**Long Video Generation**  Recent attempts combine diffusion with autoregressive modeling by varying frame noise levels (Chen et al., 2024a), but they conflict with the training paradigm of video foundation models, creating a severe train–inference gap (Kim et al., 2024) that requires costly post-training for each new foundation model (Chen et al., 2025; Teng et al., 2025). Temporal decomposition splits long videos into shorter clips to ease length and control limitations, but merging them coherently is challenging. I2V-AR generates clips autoregressively from the last frame of the previous segment, enforcing spatial continuity but lacking temporal causality, while temporal co-denoising uses overlap and synchronized denoising, which can degrade quality. Recent training-free methods, such as DiT-Ctrl (Cai et al., 2025), extend short video models by generating overlapping clips and modeling denoising trajectories to maintain temporal coherence.

**Multi-prompt Video Generation**  Multi-prompt video generation is a natural form of long-video generation. Previous works in this area have mostly focused on multi-scene video generation (Villegas et al., 2022; Cai et al., 2025), aiming for smooth scene transitions similar to video editing. Some later works (Villegas et al., 2022; Oh et al., 2024) explored using different prompts to express different actions, but they usually assign equal durations to each action and do not account for their dynamic differences. More recently, methods (Wu et al., 2025; Bansal et al., 2024; Yan et al., 2025) with time-aligned multi-action prompts have been proposed; however, these methods typically generate different video segments with different text conditions in a single denoising window, which limits the maximum video length they could generate. In this work, building on time-aligned multi-action prompts, we introduce Video Path Integral, which transforms a fixed-length video generation model into a segment-level autoregressive generation model, enabling long-term multi-action video continuation.

**Path Integral**  The path integral is one of the formulations of quantum mechanics, providing a bridge between the probabilistic behavior of the microscopic world and the deterministic patterns observed in the macroscopic world. In this paper we only borrow the underlying idea and thus give a conceptual introduction. The core notion is that a macroscopic trajectory (e.g., light traveling along the shortest path) can be viewed as the result of integrating over all possible microscopic paths, each weighted by a corresponding quantity. While every possible path is explored probabilistically, contributions reinforce along the true trajectory and cancel out elsewhere, yielding the stable macroscopic path we observe—for instance, light appearing to travel strictly along the shortest route.

## A.2  TEMPORAL ACTION BINDING

**Spatial Description Expanding**  Previous multi-prompt frameworks do not model temporal causality, relying instead on extra stitching techniques to merge independently generated clips. This creates a temporal coherence bottleneck, restricting flexible action transitions and making long videos appear as repeated edits of the same clip rather than coherent multi-action sequences. Besides, previous multi-prompt video generation frameworks also neglect to construct prompts at the action level with distinct durations. Instead, they typically assume equal time spans for all prompts, regardless of the number or type of actions involved. This mismatch often causes actions to be skipped, left incomplete, or repeated. In autoregressive generation, such omissions or incomplete executions create gaps between consecutive prompts (e.g., failing to walk to a table before being asked to pick up an item on the table), leading to error accumulation and eventual collapse of the generation.

**MLLMs as Actors for Temporal and Spatial Description Expanding**  Given a user prompt—ranging from a broad description (e.g., a man is working) to an explicit list of actions—MLLM with given contexts is employed to expand the prompt into a detailed scene description (covering environment and characters) and a coherent sequence of actions. Each action is then refined to better align with the style of pre-trained model prompts (e.g., using simple verbs with explicit motion magnitudes). Leveraging world knowledge and contextual information, the MLLM

also predicts the likely duration of each action. The result is a complete temporally grounded prompt consisting of a scene description and a sequence of temporal-action binding descriptions.

The storyboard-like prompts take the format: "name", "seed", "action-num","scene-description" for basic setup and a sequence of action descriptions "action-id" paired with "frame-num-id" follows up to bind each actions to the timeline. When different actions occur simultaneously or in close succession, we group them into a single action prompt and predict a joint duration for the whole.

## A.3 IMPLEMENTATION OF STEINSGATE

**Video Path Integral**    Video Path Integral takes a short segment of historical video as input for both spatial and temporal causality. During the generation flow, it randomly samples several historical frames as the initial frame for the TI2V model to predict multiple possible future trajectories, or video paths. It then uses weighted integration to guide these trajectories step by step, converging them along the extended direction of the historical frames—effectively "pressing play" on a paused video. We leverage the spatial-temporal disentanglement of I2V models by introducing multiple historical frames, allowing past spatial and temporal information to propagate into the continued video. This enables the generated video to be history-aware and understand temporal progression. During inference, the diffusion model is extended autoregressively—similar to block diffusion—while accurately following action sequences.

In the video continuation task, we directly take as input either user-provided videos or previously generated clips. For multi-action long video generation, we first generate the initial segment from text or image–text prompts, then apply Video Path Integral to achieve causal video continuation, producing the complete long video. The Video Path Integral process works as follows: given the length of the previous video, we select a segment as history according to a fixed ratio (note that the number of frames must satisfy the format $4N + 1$). We then initialize noise with the target length (usually the duration of the next action). During each denoising step, a random set of history frames is chosen as conditional frames. Their noisy counterparts are concatenated with the segment to be generated, after which I2V velocities are predicted from the selected history frames. Velocities corresponding to the newly generated part are combined through a weighted sum, and the result is updated according to PCG. An algorithm workflow could be refered to Alg.1.

---

**Algorithm 1** A training-free video continuation method for multi-action long video generation

---

**Input**: Pretrained video model $v_\theta$, prompts with N sgements P=$[c_{txt}, c_{img}, \{a_i\}_{i=1}^N, \{l_i\}_{i=1}^N]$, where $\{a_i\}, \{s_i\}$ are action prompts and latent frame number for each segment.
**Output**: Multi-action long videos $\mathbf{x}^{1:N}$ with action control.
 1: Generate: the first video chunk $\mathbf{z}^1$ with T2V $v_\theta(c_{txt}, a_1)$ or TI2V $v_\theta(c_{txt}, a_1, c_{img})$.
 2: Decode: Decode the video latent $\mathbf{z}^1$ into video frames $\mathbf{x}^1$.
 3: **for** each video segment $i \in [2, N]$ of multi-action long video **do**
 4:     calculate the history length $H = \lceil 0.2l_i \rceil$ and select the last $H$ frames from the previous segment $\mathbf{x}^{i-1}$ as the history frames $\{x\}_{j=1}^H$.
 5:     **for** each denoising step $t \in [1, T]$ with total denoising step $T$ **do**
 6:         **if** the denoising step $t < 0.3T$ **then**
 7:             Calculate the *monte-carlo* estimation of video path integral:
 8:             Random select $K \in \{2, 3\}$ subset $\{x_k\}_{k=1}^K$ from history frames
 9:             Calculate the weighted vector field $v_\theta(\mathbf{z}_t^i, t|\mathbf{z_h}) = \sum_{k=1}^K w_t v_\theta(\mathbf{z}_t^i, t|c_{txt}, a_i, x_k)$.
10:             Calculate the PCG vector $v_{pcg} = v_\theta(\mathbf{z}_t^i, t|\mathbf{z}_h) - v_\theta(\mathbf{z}_t^i|c_{txt}, a_i, x_K)$
11:         **else**
12:             Obtain last frame TI2V vector field $v_\theta(\mathbf{z}_t^i|c_{txt}, a_i, x_K)$;
13:             Calculate $v_\theta^*$ with PCG or CFG Guidance as in Equation 8
14:             Continuing the ODE update step: $d\mathbf{z}_t^i = v_\theta^*(\mathbf{z}_t^i, x_{1:K}, c_{txt}, a_i)dt$;
15:         **end if**
16:     **end for**
17:     Decode the video latent $\mathbf{z}_1^i$ into video frames $\mathbf{x}^i$
18: **end for**
19: **return** the multi-action long video $\mathbf{x}^{1:N}$.

---

**History Frame Selection**   we adopt a simple, empirical strategy for selecting the historical segment: we set the number of historical frames to approximately 20% of the length of the upcoming segment ($N_{\text{history}} \approx 0.2\, N_{\text{current}}$).This choice aims to balance historical conditioning with text-driven control. Since the Wan2.1 model can process at most  81 frames per inference, using too many historical frames would reduce the available capacity for generating the new segment, thereby harming text adherence. Through empirical evaluation, we find that allocating 20% history and  80% newly generated frames provides a good trade-off. Because the generation length per step varies (typically 49–81 frames), using a relative ratio is more appropriate. This results in using roughly 13–25 historical frames—sufficient to preserve temporal dynamics while avoiding excessive constraints on the upcoming motion.

the requirement above that the video length follow the 4N+1 format is imposed by the underlying pre-trained video generation model. The current Video VAE (WanVAE in our work) encodes the first frame independently and then compresses every subsequent four frames into one latent frame, which necessitates that the generated video length be of the form 4N+1. To remain aligned with the base model, we follow the same constraint: the historical segment is constructed to satisfy the 4N+1 format, and the newly generated segment follows the 4N format, so that the combined sequence (history + new frames) also conforms to the required 4N+1 structure.

During each step, we randomly sample K=2-3 history frames from a total history frame $H = N_{history}$ for a Monte-Carlo estimation for Video Path Integral. The choice of K is kept between 2–3 primarily to control the per-step inference cost. Using larger K would make the sampling time grow quickly and become impractical. The specific frames selected may vary across steps, and after multiple denoising iterations, nearly all historical frames are eventually covered. This Monte Carlo estimation strategy allows the otherwise costly process of mixing H I2V paths to be amortized across iterations, improving sampling efficiency in practice.

**Guidance Interval**   Guidance Interval refers to the use of Video Path Integral during the first part of the denoising process, specifically from time step t = 0 to t = 0.3 (equivalent to the first 15 discrete steps under a 50-step DDIM schedule). This interval was chosen as a balance between efficiency and performance: applying Video Path Integral over more steps can slightly improve results, but the sampling time increases linearly with the number of steps, making larger thresholds less cost-effective.

**Path Convergence Guidance**   Similar to Classifier-Free Guidance and AutoGuidance, weak-to-strong guidance methods generally require an extrapolative formulation, i.e., using a guidance weight larger than 1. The key intuition is that interpolation between the weak and strong directions often leads to worse results than using the strong direction alone, whereas extrapolation shifts further along the 'weak-to-strong' direction $v_{\text{strong}} - v_{\text{weak}}$ (as shown in Fig. 4c), typically yielding outputs better than using the strong direction alone.

Theoretically, this can be understood from the perspective of distribution shift: the guided direction corresponds to a modified distribution $P(x)P(\text{good} \mid x)^w$. Only when the exponent w > 1 does the distribution shift sufficiently toward the desired region, enabling effective guidance.

A.4   EXPERIMENT SETUP

**Baselines**   We implement all Causality Enforcing Baselines for causal video continuation based on the same underlying model, Wan2.1, and ensure that these baselines share the Temporal Action Binding framework with SteinsGate, i.e., they are conditioned on the same multi-action prompts. For SDEdit, we adopt a similar idea by replacing the historical segment of the generated video during sampling with a noised version of the real history. For RG-Flow, we interpolate between the generated history and the ground-truth history, and then predict an updated vector field based on the interpolated history:

$$\hat{v}_{\mathbf{z}}(\mathbf{z}_t|\mathbf{z}_h) = \frac{\mathbb{E}_p[\mathbf{z}_1|\mathbf{z}_t|\mathbf{z}_h] - \mathbf{z}_t}{1-t}, \; \mathbb{E}_p[\mathbf{z}_1|\mathbf{z}_t|\mathbf{z}_h][1:H] = (1-t)\mathbf{z}_h + t\mathbb{E}_p[\mathbf{z}_1|\mathbf{z}_t][1:H] \quad (9)$$

Finally, I2V-AR simply generates the subsequent video segment by using the last frame of the previous segment as input to the TI2V model.

We select representative baselines for multi-action long video generation. DiTCtrl and FIFO-Diffusion is the most related training-free baselines for multi-prompt video generation. Meanwhile, SkyReel-v2 and MAGI-1 are indeed the large-scale training-based video continuation models, which we also include for comparison. Furthermore, we include the commercial model Sora (storyboard enhanced version) for a system level comparison. Results suggest that SteinsGate achieve better performance than all training-free baselines and comparable performance with large-scale training-based baselines and commercial models. SteinsGate, as a training-free proof of concept, demonstrates both the feasibility and the advantages of the InstructVC approach—combining global causal planning with local video-causal continuation for temporal causality modeling. It also shows the practical value of Video Path Integral as a novel form of temporal guidance.

All baselines are experimented following their official Hugging Face repositories or codebases. Experiments are conducted on a single NVIDIA A100 GPU.

**Setups** We build SteinsGate on Wan2.1, a leading open-source Text-Image-to-Video DiT model. For multi-prompt generation, we apply Temporal Action Binding to structure prompts, setting each clip length to its predicted duration at 15 fps. For single-prompt generation, we concatenate the scene description and all action prompts, with the total length set to the sum of action durations. All prompts are expanded with GPT-4o (Hurst et al., 2024), generation uses an Euler sampler with 50 steps, and outputs are rendered at 480×720 resolution.

All baselines are evaluated on the same single NVIDIA A100 80GB GPU. For video continuation baselines, we re-implemented each method within a unified codebase to ensure consistent samplers and denoising steps. For multi-action long-video generation, we conduct a system-level comparison: all baselines run with their default, recommended configurations, with no inference-time constraints. We compare only the final video quality. For latency, the vanilla I2V-AR baseline (no extra inference-time overhead, just for video generation) takes approximately 30 minutes to generate a 15-second video, while SteinsGate requires around 38 minutes, an acceptable 25% inference time overhead in practice.

**Metrics** We adopt the evaluation metrics used in DiTCtrl [1] and VBench [3], including CSCV, Motion Smoothness, and Text-Image Alignment. The detailed definitions can be found in the original paper; here we provide a brief summary:

1. Clip Similarity Coefficient of Variation (CSCV) (Cai et al., 2025): a metric specifically designed to evaluate the transition smoothness of multi-prompt videos, defined as:

$$s_i = x_i^T x_{i+1}, i = 1, \ldots, N - 1, \text{CSCV} = \frac{1}{1 + \lambda \frac{\sigma(s)}{\mu(s)}} \tag{10}$$

where $x_i$ denotes clip frame features, $\sigma$ and $\mu$ are standard deviation and average for clip similarity score respectively. The Coefficient of Variation $CV = \sigma(s)/\mu(s)$ describes the degree of uniformity, which can largely punish the isolated situation. The function $\frac{1}{1+\lambda()}$ projects the score to [0,1], the larger the better.

2. Text-Image Alignment: a commonly adopted metric using CLIP Similarity (Cai et al., 2025) to assess the alignment between given prompts and output video clips

3. Motion smoothness: a metric from VBench (Huang et al., 2024) to evaluate whether the motion in the generated video is smooth and follows the physical law of the real world.

Since our focus is on evaluating motion smoothness during multi-prompt transitions—an aspect closely tied to video continuation quality—it is difficult to establish a consistent and objective standard across different human raters, which can easily introduce bias. Given that we do not have the resources to recruit and train professional annotators, we instead use automated metrics such as the Clip Similarity Coefficient Variation (CSCV), which offer consistent measurement and reliably reflect multi-prompt transition quality.

**Benchmark Construction** Using GPT-4o, we constructed a diverse prompt dataset consisting of 60 long-form prompts with character and scene descriptions, along with a set of action descriptions. Since our method is training-free, the dataset is used purely for testing and contains no train/val

| Metric | PCG Weight | | | History Length Ratio | | | Guidance Interval | | | Selected Frame Num. | | |
| --- | --- | --- | --- | --- | --- | --- | --- | --- | --- | --- | --- | --- |
| | w=0.5 | w=1.0 | w=1.5 | r=10% | r=20% | r=30% | $t_{\text{mid}}$=0.1 | $t_{\text{mid}}$=0.3 | $t_{\text{mid}}$=0.5 | K=1 | K=2 | K=3 |
| CSCV | 0.75 | 0.78 | 0.82 | 0.75 | 0.82 | 0.79 | 0.76 | 0.82 | 0.81 | 0.74 | 0.79 | 0.82 |
| Motion Smoothness | 0.93 | 0.95 | 0.97 | 0.94 | 0.97 | 0.95 | 0.95 | 0.97 | 0.97 | 0.94 | 0.96 | 0.97 |
| Text-Image Alignment | 0.31 | 0.29 | 0.32 | 0.31 | 0.32 | 0.31 | 0.31 | 0.32 | 0.33 | 0.31 | 0.31 | 0.32 |

Table 2: More Ablations. History Length Ratio specifies how many frames from the end of the previous segment are used as history, computed relative to the length of the upcoming segment; it balances the amount of historical context versus newly generated content within a fixed window. $t_{\text{mid}}$ determines the cutoff in the denoising schedule after which we stop applying Video Path Integral and instead use the vector field conditioned on the last frame. Selected Frame Number denotes how many historical frames are sampled in each step when applying the Video Path Integral's Monte-Carlo estimation; these frames are used to compute the I2V video paths for that step.

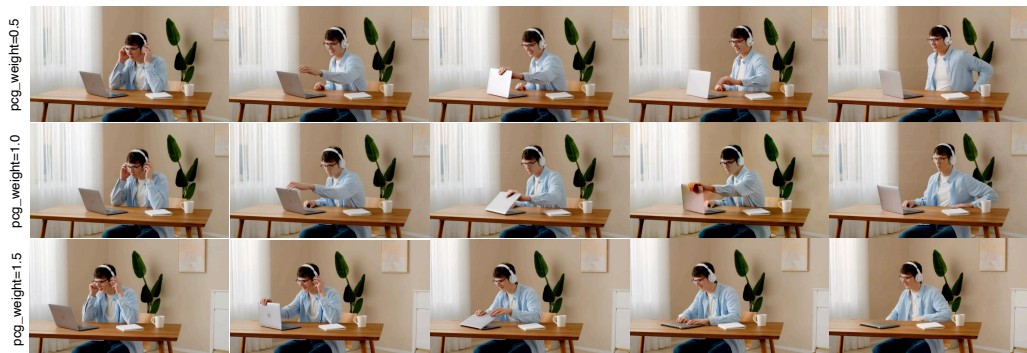

Figure 8: Qualitative Ablation Study for Path Convergence Guidance weight. Prompt: "a man working at a desk and then close his computer for a leave."

splits. Most prompts are adapted from the MinT (Wu et al., 2025) test set via GPT-4o rewriting, while a smaller portion is directly generated by GPT-4o.

Roughly 30% of the test prompts contain four actions, about 50% contain three actions, around 10% contain more than four actions, and the remaining 10% contain two actions. The scenes are primarily human-centric (around 80%), which aligns with the strengths of current video generation models and the typical application setting for multi-action long-video generation. These include outdoor scenes, indoor scenes, full-body and half-body shots, as well as various human–object interaction scenarios. The remaining 20% involve other scene types such as animals and landscapes. In terms of duration, about 70% of the videos are around 20 seconds long, 18% are around 15 seconds, and the remaining 12% exceed 20 seconds.

## A.5 MORE EXPERIMENTS RESULTS

**More Ablations** We conducted additional ablation studies, and the results are shown in Table 2. The findings indicate that our choice of history ratio r = 20% achieves a good balance among the model's maximum manageable temporal window, the difficulty of history adherence, and consistency with both text and historical context. For each step of the Video Path Integral, we set K = 2, which offers strong performance while keeping the inference-time overhead acceptable. For Path Convergence Guidance, we adopt the more effective extrapolation strategy with w = 1.5. For the Guidance Interval, we use $t_{\text{mid}} = 0.3$, which provides the best trade-off between inference-time cost and generation quality.

We also provide more qualitative results in Figure 8. The Monte-Carlo–estimated Video Path Integral (VPI) velocity exhibits high variance and contains multiple plausible motion directions, which can be observed from the mixed and unstable motions when pcg = 1.0. When pcg < 1.0, the model interpolates between the VPI velocity and the last-frame I2V velocity, causing it to lean toward the latter. Although this reduces motion mixing, it often leads to incorrect motion directions due

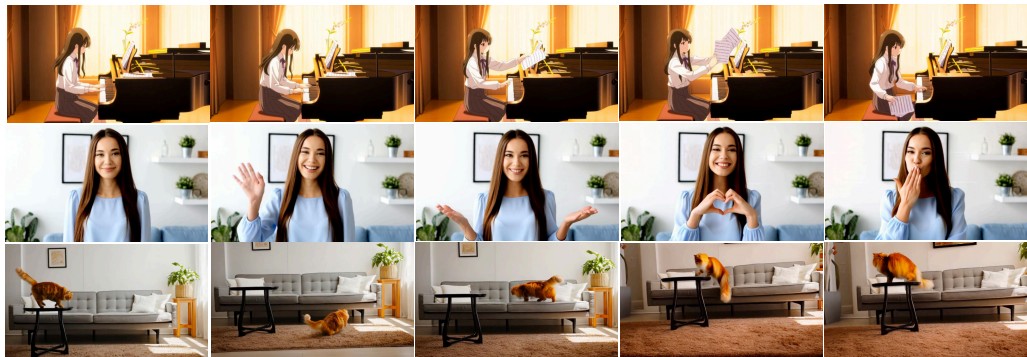

Figure 9: More qualitative cases. Prompt for the upper: "Anime style. A girl sits at a piano playing.", the middle: "A woman greets and talks to the camera, then ends by making a heart gesture and blowing a kiss toward the camera.", the bottom: "A cat jumps off a table onto the floor, then leaps onto a sofa, and finally walks back to the table."

to over-reliance on the last-frame guidance. In contrast, when pcg $> 1.0$, the model extrapolates between the two velocities, effectively suppressing the potentially incorrect direction suggested by the last-frame I2V velocity. As a result, the guidance shifts the model toward more plausible and coherent motion directions.

More qualitative results including human, animate-style and animal are provided in Figure 9.

## A.6 USE OF LLMs

**Manuscript preparation.** We used a large language model only for polishing the paper and all contents and primary writing were conducted by the authors.

**Method role of MLLM.** In our method, a multi-modal large language model (MLLM) serves as the executor of Temporal Action Binding. The instruction prompts and in-context learning examples used to train this behavior are provided below.

---

Please help me enrich the user prompt for video generation. Given a single text prompt, you need to extend it to a list of temporal captions and a global caption. Or given a list of temporal captions, you need to give a global caption and polish the temporal captions with reference to the captions before. The goal is to enhance the short user prompt and provide more context to the video generator, so that the generated video contains more detailed and coherent events. Please focus solely on visual elements and actions without mentioning ambient sounds or other non-visual sensory details.

The temporal captions describe sequential events happening in the scene and their lasting time (in frame_num). It should follow these rules:

1. Each event should maintain similar entities and background scenes.
2. Each event prompt must contain only a single motion or action.
3. Each event prompt can be easily described by a video clip shorter than 5s (81frames).
4. The event lasting time is measured in frame numbers (the default fps=15). It should reflect the time needed to perform each action in real world which is usually different between events. The frame number should be in 4*N+1 format.
5. Each event should be smoothly connected to its adjacent events, i.e., they can be plausibly presented in a video without any cuts.
6. Each event prompt can also contain the camera motion at the beginning if it is important to the event.
7. There should be no more than 5 events.
8. The whole video should not exceed 30s.
9. Be careful not to alter the action sequence in the provided temporal captions; ensure that the action sequence remains consistent with the original temporal captions.
10. Refine temporal captions to ensure smooth narrative flow by incorporating contextual details from the global caption, such as spatial layout and object relationships. Use appropriate pronouns to maintain consistent reference and create seamless transitions between consecutive temporal captions. Objects should not appear suddenly—if referenced in later captions, they must be introduced or implied in preceding ones. Briefly mention results of previous actions in subsequent captions, such as adding "with the VR headset still on" after it was put on earlier. The combined word count of the global caption plus any single temporal caption should be 80-100 words.

The global caption is a general description of the scene containing:

1. The background of the scene such as the spatial layout of objects.
2. The main entities involved in the scene and their attributes, such as clothing, age, and appearance of a person.
3. The weather if it is an outdoor scene.
4. Camera angles and movements if they are important to the scene.
5. Do not include an overall summary of the video content; focus on describing the appearance of the environment and the characters.
6. You may focus on describing appearance, but avoid introducing complex spatial layouts or excessive number of objects.

Example 1-N: ...... (in context learning examples)

---

