# OpenReview forum: "SteinsGate: Adding Causality to Diffusions for Long Video Generation via Path Integral"
_ICLR.cc/2026/Conference — ICLR 2026 Poster_

### Official Review · Reviewer_inkc · 2025-10-28

**Soundness:** 3
**Presentation:** 3
**Contribution:** 3
**Rating:** 8
**Confidence:** 4

**Summary:**

The paper introduces Instruct-Video-Continuation (InstructVC), a two-stage inference-time framework for multi-action long video generation that adds temporal causality to pretrained video diffusion models. Stage 1 (Temporal Action Binding) uses an MLLM with in-context examples to decompose and expand a user prompt into a scene description plus a sequence of action-duration pairs. Stage 2 (Causal Video Continuation) converts an Image-to-Video (I2V) diffusion model into a history-aware autoregressive continuer via Video Path Integral (VPI), which integrates multiple I2V “video paths” from sampled historical frames to bias future trajectories toward history-consistent directions. The practical instance SteinsGate adds three optimizations—Guidance Interval, History-aligned Redistribution, and Path Convergence Guidance—to reduce compute and improve alignment. Experiments on a constructed InstructVC benchmark and ablations report improved temporal control, motion smoothness, and multi-action continuity vs. several inference-time baselines and show competitive performance with heavier training-based methods.

**Strengths:**

1. The paper isolates two well-motivated, complementary challenges for long video generation—temporal causality and temporal control—and proposes a coherent two-stage solution.

2. Video Path Integral provides a principled, interpretable way to propagate temporal information from multiple history frames and to bias generation toward causally consistent continuations without retraining the foundation model.

3. The three SteinsGate optimizations (Guidance Interval, History-aligned Redistribution, Path Convergence Guidance) address computational cost and estimation noise, making the method more viable in practice.

4. Temporal Action Binding leverages MLLM capabilities for temporally grounding and duration prediction, enabling fine-grained temporal control over multi-action sequences.

5. The paper includes qualitative continuation examples, a system-level comparison for multi-action long generation, and ablations that quantify contribution of each component.

6. The method can be applied at inference-time to pretrained TI2V/DiT models, which increases practical applicability and lowers barrier for adoption.

**Weaknesses:**

1. The VPI argument is mostly conceptual and analogy-driven (path integral intuition); formal bounds or rigorous analysis of when/why VPI converges to the true conditional distribution are missing.

2. Performance and long-term consistency appear sensitive to how much and which historical frames are used, but the paper provides only heuristic choices and limited analysis of failure modes for varying history lengths.

3. Although in-context learning is used to reduce implausible decompositions, reliance on an MLLM can still produce action sequences or durations that are out-of-distribution for the pretrained video model; mitigation and quantitative assessment are limited.

4. Benchmark construction and evaluation transparency: the InstructVC benchmark is constructed by expanding dense-caption datasets, but details on dataset size, prompt diversity, split protocols, evaluation metrics definitions, and human evaluation procedures are sparse.

5. Reported baselines are appropriate, but the paper lacks comparisons against recent large-scale training-based long-video systems (or more thorough hyperparameter-matched baselines) that could better contextualize headroom and limitations.

6. The metrics focus on smoothness and CLIP alignment; deeper semantic correctness (action completeness, causal correctness judged by humans) and statistical significance of improvements are not fully reported.

**Questions:**

1. Can the authors precisely define the notation and dimensionality used in Eq. (5) for Video Path Integral, including what each index (i, j, K, H) represents and how histories are sampled in practice?

2. How exactly is the mapping performed from image-conditioned I2V vector fields to the history-conditioned vector fields ve(Zt | xj); give the algorithmic steps used at each denoising iteration.

3. What is the precise schedule for Guidance Interval: is it a fixed fraction of the continuous time variable, and how was 0.3 chosen?

4. When you say the Video Path Integral approximates p(Zi | Zh) by product of frame-wise conditionals (Eq. 6), what independence assumptions are implied and when do they break down?

5. How many prompts and total videos are in the InstructVC benchmark (train/val/test splits), and what is the distribution of number-of-actions, scene types, and durations?

6. What are the exact definitions and implementations for CSCV+, Motion Smoothness, and Text-Image Alignment metrics, and how were thresholds or evaluation pipelines calibrated?

7. Were human raters used for causal correctness or action completeness? If so, how many raters, what instructions, and inter-rater agreement (e.g., Cohen’s kappa) were measured?

8. For quantitative comparisons, were compute budgets and sampling steps matched across methods (same sampler, same number of denoising steps)? Please provide latency and GPU-memory numbers.

9. Please provide a sensitivity analysis for the number of history frames K, the selection strategy for those frames, and the history-length ratio used for different action durations.

10. How robust is SteinsGate to noisy or incorrect history (e.g., partially corrupted frames, temporal jitter, or mismatched last-frame poses)?

11. How does performance degrade over very long sequences beyond 30s, and what empirical evidence supports the claim that VPI reduces error accumulation compared to I2V-AR?

12. What failure modes are most common (motion reversal, skipped actions, object disappearance), and can you quantify their frequencies per baseline?

13. What MLLM model/version and temperature/prompting strategy was used for Temporal Action Binding, and what in-context examples were provided (exact prompts/examples)?

13. Please provide quantitative measures of MLLM output quality: rate of hallucinated or physically implausible action sequences, distribution shift vs. the TI2V training captions, and any post-filtering applied.

14. Will code, exact prompts, model checkpoints (Wan2.1 config and weights), and InstructVC benchmark splits be released?

---

> ### Author Response · Authors · 2025-11-21
> **Thanks for your careful review and recognition! (1/4)**
>
> Thanks for your warm words and recognition! We have updated the appendix according to your suggestions (Marked by blue). Sections with significant changes are marked by coloring their titles blue, while the content remains uncolored for clarity. We would like to address your concerns as follows:
>
> **W1, Q1, Q4: Detailed Definition and Further Analysis on Video Path Integral**
>
> We apologize for the confusion. We will add clarifications on the relevant index notation and provide additional analysis in the revised manuscript.
>
> **Detailed Definition of Eq.(5)**
>
> z_t denotes the noisy video latent being generated at time step t, with shape (N, h, w, c), where N, h, w represent the number of frames and the spatial height and width in the compressed WanVAE latent space, and c=48 is the fixed VAE channel size. $z_h$ = (H, h, w, c) represents the historical latent frames from the previous chunk, where H = 0.2N, i.e., roughly 20% of the frames in $z_t$ (details discussed below).
>
> At each step t, a small subset of K historical frames where single frame $x$=(1, h, w, c), is randomly sampled, typically $K \in \{2,3\}$. The specific frames selected may vary across steps, and after multiple denoising iterations, nearly all historical frames are eventually covered. This Monte Carlo estimation strategy allows the otherwise costly process of mixing H I2V paths to be amortized across iterations, improving sampling efficiency in practice.
>
> Here, indices i and j refer to the index of frames in the full historical frame list and the sampled subset within a single iteration, respectively.
>
> **Further Analysis on Video Path Integral**
>
> In Equation 6, $p(z_t \mid z_h) \propto \prod_{j=1}^H p(z_t \mid x_j)$
>
> the key underlying assumption is that each single-frame conditional distribution can be expressed in **exponential-family form**: $P(z_t \mid x_j) \propto \exp(f(z_t, x_j)).$
>
> Under this assumption, the product distribution becomes:
>
> $\prod_{j=1}^H p(z_t \mid x_j) \propto \exp\Big(\sum_{j=1}^H f(z_t, x_j)\Big).$
>
> If the joint conditional distribution can be written as: $p(z_t \mid z_h) \propto \exp\Big(F(z_t, z_h)\Big),$
>
> and F can be decomposed as a sum of single-frame contributions:
>
> $F(z_t, z_h) = \sum_{j=1}^H f(z_t, x_j),$
>
> then the proportional relationship in Equation 6 holds, i.e., the joint distribution is **proportional** to the product of the single-frame conditionals without requiring exact normalization.
>
> Importantly, distributions estimated in the **diffusion family** (including flow matching) satisfy this form, where the energy function f is parameterized by a neural network. This justifies the proportional decomposition in Equation 6.
>
> Overall, when the energy function F, parameterized by a neural network, can correctly capture the contribution of each individual historical frame, i.e., after path mixing, the network can generalize to the distribution satisfying that any single historical frame is the given frame, then the combination of single-frame conditionals can accurately approximate the joint historical conditional distribution. Under this assumption, **Video Path Integral** is able to converge.
>
> In practice, since we use Monte Carlo sampling, if the number of sampled frames is insufficient, they may fail to represent the full set of historical frames. In addition, if the historical dynamics are highly complex and exceed the network’s generalization capacity, the estimation may also become inaccurate.
>
> **Q2-Q3, Q15:  Implementation details and algorithm workflow**
>
> We apologize for the confusion. **We have provided additional implementation details and pseudocode for the algorithm workflow in the Appendix A.3,** and the paper will also be open-sourced after acceptance.
>
> **Guidance Interval** refers to the use of **Video Path Integral** during the first part of the denoising process, specifically from time step t = 0 to t = 0.3 (equivalent to the first 15 discrete steps under a 50-step DDIM schedule). This interval was chosen as a balance between efficiency and performance: applying Video Path Integral over more steps can slightly improve results, but the sampling time increases linearly with the number of steps, making larger thresholds less cost-effective.

---

> ### Author Response · Authors · 2025-11-21
> **Thanks for your careful review and recognition! (2/4)**
>
> **W3, Q13-14: Reliance on MLLM**
>
> Improvements and evaluation of MLLMs are not the focus of this paper, as they represent an orthogonal research direction. Running evaluation for MLLM requires long-video generation for each generated prompt, which is extremely time-consuming and labor-intensive. Thus Our primary evaluation emphasis is instead on the quality of multi-action long-video generation under fixed generated prompts. In this work, we use GPT-4o for MLLM evaluation without any post-filtering. Moreover, because the training data of the pre-trained model Wan2.1 is not publicly available, it is difficult to assess the distributional gap between MLLM-generated prompts and captions from the training set. Therefore, we choose to conduct a **system-level comparison**, focusing primarily on the final **multi-action long-video generation quality**.
>
> The meta-prompt could be found in the Appendix A.4. Some example of in-context prompts are listed as follows：
>
> Example 1:
>
> Input: “A man in the gym shows how to do exercises.”
>
> Output (In json format):
> ```json
> {
>       "name": "A man in the gym",
>       "seed": 42,
>       "block_num": 3,
>       "global_text": "A low angle, full shot from a static camera of a dark-skinned man in the gym. The gym has a white tile ceiling, light grey walls with mirrors, and a grey floor. On the ceiling is a yellow ventilation tube. In the background is a grey-yellow training equipment.",
>       "action1": "A static camera shows the man is leaning on the horizontal bar and speaks to a camera with a bottle in his hand.",
>       "frame_num_1": 65,
>       "action2": "The camera zooms in showing the man walking back with a bottle in his hand.",
>       "frame_num_2": 81,
>       "action3": "The camera pans left showing the man puts a bottle on the floor and takes up a dumbbell.",
>       "frame_num_3": 81
> }
> ```
>
> Example 2:
>
> Input: “A man plays with a VR headset with excited reactions.”
>
> Output (In json format):
> ```json
> {
>      "name": "A man plays with a VR headset with excited reactions.",
>      "seed": 42,
>      "block_num": 4,
>      "global_text": "A medium shot from a handheld camera shows a fair-skinned man standing in a living room. The camera pans to the right and left and dollies backward and forward. He has black hair, a beard, and a mustache. He wears a horizontally green-white striped t-shirt, beige trousers, and black thread on his left wrist. In the background are a light brown couch and a dark glass window with wooden frames, a lamp on the right side, several green plants, and a stand on the left with several items held like a white round clock, books, a photo frame, and a green plant in a white pot.",
>       "action1": "A man wearing a pair of glasses holds the VR headset in his hands.",
>       "frame_num_1": 33,
>       "action2": "The man places the VR headset on his head.",
>       "frame_num_2": 49,
>       "action3": "The man lowers his hands and looks around with the VR headset on his head.",
>       "frame_num_3": 65,
>       "action4": "The man tilts his hands left and right and smiles with the VR headset on his head.",
>       "frame_num_4": 81
> }
> ```
> Example 3:
>
> Input:
>
> [0.0s → 1.4s]: The woman waves with her right hand.
>
> [1.4s → 3.6s]: The woman turns her right hand to a thumbs-up gesture.
>
> [3.6s → 6.4s]: The woman makes a heart gesture with both hands.
>
> [6.4s → 9.1s]: The woman gives a blow kiss with her right hand.
>
> Output (In json format):
>
> ```json
> {
>     "name": "a woman in front of a camera",
>     "seed": 37,
>     "block_num": 4,
>     "global_text": "A medium shot from an eye-level static camera captures a fair-skinned young woman with long black hair standing in a cozy, softly lit room. She wears a pink sweater and light makeup. Behind her is a beige wall with a small potted plant and a wooden shelf holding decorative items. The woman engages directly with the camera while maintaining eye contact and a friendly smile throughout.",
>     "action1": "The woman waves with her right hand, smiling at the camera.",
>     "frame_num_1": 33,
>     "action2": "The woman smoothly transitions her right hand into a thumbs-up gesture.",
>     "frame_num_2": 33,
>     "action3": "The woman brings both hands together in front of her chest to form a heart gesture.",
>     "frame_num_3": 65,
>     "action4": "The woman lifts her right hand to her lips and gives a blow kiss toward the camera.",
>     "frame_num_4": 81
> }
> ```

---

> ### Author Response · Authors · 2025-11-21
> **Thanks for your careful review and recognition! (3/4)**
>
> **Q1, W2, Q9: History Frame Selection**
>
> Selecting historical frames is inherently challenging. Ideally, the choice should depend on understanding the video content, and determining optimal video contexts is itself an independent research problem. The core contribution of our work, however, is to provide a **training-free proof-of-concept** for the InstructVC approach to long-video generation. Accordingly, our focus is on demonstrating **how to convert a pretrained video generation model into a video continuation model**, endowing a purely bidirectional diffusion model with the ability to handle **variable-length contexts**, rather than on designing a dedicated historical-frame selection mechanism.
>
> Therefore, we adopt a simple, empirical strategy for selecting the historical segment:
>
> we set the number of historical frames to approximately **20% of the length of the upcoming segment** $(N_{\text{history}} \approx 0.2\, N_{\text{current}})$. **This choice aims to balance historical conditioning with text-driven control.** Since the Wan2.1 model can process at most ~81 frames per inference, using too many historical frames would reduce the available capacity for generating the new segment, thereby harming text adherence. Through empirical evaluation, we find that allocating ~**20% history and ~80% newly generated frames** provides a good trade-off. Because the generation length per step varies (typically 49–81 frames), using a relative ratio is more appropriate. This results in using roughly 13–25 historical frames—sufficient to preserve temporal dynamics while avoiding excessive constraints on the upcoming motion.
>
> During each step, we randomly sample K=2-3 history frames from a total history frame $H=N_{history}$ for a Monte-Carlo estimation for Video Path Integral. The choice of K is kept between 2–3 primarily to control the per-step inference cost. Using larger K would make the sampling time grow quickly and become impractical. The specific frames selected may vary across steps, and after multiple denoising iterations, nearly all historical frames are eventually covered. This Monte Carlo estimation strategy allows the otherwise costly process of mixing H I2V paths to be amortized across iterations, improving sampling efficiency in practice.
>
> **Q10-12: Failure Mode and Robustness analysis**
>
> **Failure Mode Analysis**
>
> Without **Temporal Action Binding** (i.e., when performing chunk-by-chunk video continuation under a fixed global prompt), all methods tend to exhibit **action repetition** or **action omission**. This happens because a video generator naturally tries to map the text prompt to the currently generated clip: each chunk tends to reproduce the **earlier major actions** (leading to repetition), while **later actions** may be ignored due to the limited temporal span of a single chunk. Temporal Action Binding is essential to mitigate such instability in multi-action generation.
>
> With our proposed Temporal Action Binding, the main failure mode is a degradation in Transition Smoothness  between multiple-prompt videos. A more noticeable subset of failures involves action direction reversals or action conflicts. These typically occur for actions that are directionally symmetric—such as *pick up vs. put down* or *open vs. close*. When only the final frame of an action is observed, these pairs can be difficult to distinguish, making the model more prone to triggering such direction-reversal or conflict failure modes.
>
> Therefore, we primarily rely on the **CSCV** metric to evaluate Temporal Smoothness during multi-prompt transitions, along with overall **Motion Smoothness** and **Text-Image Alignment**. As shown in Table 1, SteinsGate significantly improves Temporal Smoothness and achieves the best performance among all training-free methods, even reaching a level comparable to recent large-scale pre-trained models such as SkyReel-V2 and MAGI-1.
>
> **Robustness Analysis**
>
> Building on our proposed Temporal Action Binding, our **SteinsGate** method demonstrates **stronger dynamic understanding and higher stability** compared to baselines such as I2V-AR with Temporal Action Binding (w/o VPI setting in Table 1) . Moreover, by referencing **multiple historical frames**, SteinsGate benefits from an averaging effect: degradation in any single historical frame can be compensated by others, resulting in a more comprehensive understanding of both the **temporal dynamics** and **spatial context** of the history frames. As discussed earlier, the robustness of SteinsGate also depends on the foundation model’s generalization ability. As base models continue to improve, the robustness of SteinsGate will further increase, leading to even better performance.

---

> ### Author Response · Authors · 2025-11-21
> **Thanks for your careful review and recognition! (4/4)**
>
> **W4-6, Q5-8: Experiment Setups and Benchmark Construction**
>
> Sorry for the confusion. We have included the details of the benchmark in the appendix, including the definitions of evaluation metrics.
>
> **Experiment Setups**
>
> **Metrics:** We adopt the evaluation metrics used in DiTCtrl [1] and VBench [3], including CSCV, Motion Smoothness, and Text-Image Alignment. The detailed definitions can be found in the original paper; here we provide a brief summary:
>
> - Clip Similarity Coefficient of Variation (CSCV) [1]: a metric specifically designed to evaluate the *transition smoothness* of multi-prompt videos, defined as:
>
>      $s_i=x_i^Tx_{i+1}, i=1, …, N-1,\ CSCV=\frac{1}{1+ \lambda\frac{\sigma(s)}{\mu(s)}}$
>
>     where $x_i$  denotes clip frame features, σ and µ are standard deviation and average for clip similarity score respectively. The Coefficient of Variation CV= σ(s)/µ(s) describes the degree of uniformity, which can largely punish the isolated situation. The function $\frac{1}{1+λ(·)}$ projects the score to [0,1], the larger the better.
>
> - Text-Image Alignment: a commonly adopted metric using CLIP Similarity [2] to assess the alignment between given prompts and output video clips
> - Motion smoothness: a metric from VBench [3] to evaluate whether the motion in the generated video is smooth and follows the physical law of the real world.
>
> Since our focus is on evaluating motion smoothness during multi-prompt transitions—an aspect closely tied to video continuation quality—it is difficult to establish a consistent and objective standard across different human raters, which can easily introduce bias. Given that we do not have the resources to recruit and train professional annotators, we instead use automated metrics such as the Clip Similarity Coefficient Variation (CSCV), which offer consistent measurement and reliably reflect multi-prompt transition quality. We kindly ask for your understanding.
>
> **Baselines:** We select representative baselines for multi-action long video generation. DiTCtrl and FIFO-Diffusion is the most related training-free baselines for multi-prompt video generation. Meanwhile, SkyReel-v2 and MAGI-1 are indeed the large-scale training-based video continuation models, which we also include for comparison. Furthermore, we include the commercial model Sora (storyboard enhanced version) for a system level comparison.  Results suggest that SteinsGate achieve better performance than all training-free baselines and comparable performance with large-scale training-based baselines and commercial models. SteinsGate, as a training-free proof of concept, demonstrates both the feasibility and the advantages of the InstructVC approach—combining global causal planning with local video-causal continuation for temporal causality modeling. It also shows the practical value of Video Path Integral as a novel form of temporal guidance.
>
> All baselines are evaluated on the same single NVIDIA A100 80GB GPU. For video continuation baselines, we re-implemented each method within a unified codebase to ensure consistent samplers and denoising steps. For multi-action long-video generation, we conduct a system-level comparison: all baselines run with their default, recommended configurations, with no inference-time constraints. We compare only the final video quality. For latency, the vanilla I2V-AR baseline (no extra inference-time overhead, just for video generation) takes approximately 30 minutes to generate a 15-second video, while SteinsGate requires around 38 minutes, an acceptable ~25% inference time overhead in practice.
>
> **Benchmark Construction**
>
> Using GPT-4o, we constructed a diverse prompt dataset consisting of 60 long-form prompts with character and scene descriptions, along with a set of action descriptions. Since our method is training-free, the dataset is used purely for testing and contains no train/val splits. Most prompts are adapted from the MinT [4] test set via GPT-4o rewriting, while a smaller portion is directly generated by GPT-4o.
>
> Roughly 30% of the test prompts contain four actions, about 50% contain three actions, around 10% contain more than four actions, and the remaining 10% contain two actions. The scenes are primarily human-centric (≈80%), which aligns with the strengths of current video generation models and the typical application setting for multi-action long-video generation. These include outdoor scenes, indoor scenes, full-body and half-body shots, as well as various human–object interaction scenarios. The remaining ~20% involve other scene types such as animals and landscapes. In terms of duration, about 70% of the videos are around 20 seconds long, 18% are around 15 seconds, and the remaining ~12% exceed 20 seconds.
>
> [1] Minghong Cai, et al. Ditctrl CVPR 2025
>
> [2] Jack Hessel, et al. Clipscore arxiv
>
> [3] Ziqi Huang et al. Vbench CVPR 2024
>
> [4] Ziyi Wu et al. Mind the time: Temporally-controlled multi-event video generation. In *CVPR*, 2025

---

### Official Review · Reviewer_8cn5 · 2025-11-01

**Soundness:** 3
**Presentation:** 2
**Contribution:** 2
**Rating:** 4
**Confidence:** 4

**Summary:**

This paper proposes a framework for multi-action long video generation, Instruct-Video-Continuation (InstructVC) that contains two control stages: Temporal Action Binding and Causal Video Continuation. The authors further introduce an inference-time instance SteinsGate.

**Strengths:**

1. The paper is well organized.

2. The topic of long video generation is worth exploring in the research community. This paper targets this important problem.

3. The paper provides some video demos to help reviewers better evaluate the performance of the proposed method.

**Weaknesses:**

There are some concerns and questions about this paper:

1.	The third paragraph of the introduction is a bit confusing. The first part mentions two methods for generating long videos: temporal expanding and temporal decomposition. So, to which category does the latter part of the paragraph, I2V-AR, and another work, belong?

2.	It is recommended that the author provide a corresponding video for Fig. 1. A few frames are insufficient to convey the author's intended meaning.

3.	The author repeatedly emphasizes the term "temporal causality." What exactly does this term mean? Why does its absence lead to phenomena such as temporal inconsistency and action direction conflict in the generated video?

4.	The two paragraphs following the introduction cover a wide range of topics, such as Temporal Action Binding, Causal Video Continuation, Guidance Interval, History-aligned Redistribution, and Path Convergence Guidance. This can be quite confusing, as it raises the question: which part is the core of the proposed method? Which part is the key to solving long-action video generation?

5.	The video demos in the supplementary materials all seem to involve very simple actions and are basically long videos within a single content. I think focusing on story-based long video generation would be better. Additionally, the video "woman_gestures" is clearly discontinuous, exhibiting obvious splicing artifacts from multiple video clips.

**Questions:**

Please see above.

---

> ### Author Response · Authors · 2025-11-21
> **Thank you for your careful review (1/2)**
>
> Thank you for your careful review! We have updated the introduction, appendix and supplementary according to your suggestions (Marked by blue). Sections with significant changes are marked by coloring their titles blue, while the content remains uncolored for clarity. We would like to address your concerns as follows:
>
> **W3: The definition and impact of Temporal Causality**
>
> **Definition of Temporal Causality**
>
> We apologize for the confusion, and we will explicitly define this concept in the paper. Our definition of Temporal Causality follows the standard notion used in autoregressive sequence modeling: when a sequence is factorized along the sequential dimension via the chain rule, **the conditional dependency on previous segments is referred to as causality.** In the context of long-video generation, this means that after dividing a long video into consecutive temporal segments, the dependency of each segment on the previous ones constitutes **Temporal Causality**.
>
> More formally, when a long video is divided into sequential chunks along the temporal dimension and factorized using the chain rule, we obtain $p(z_{1:N}) = \prod_{i=1}^{N} p(z_i \mid z_{<i}).$
>
> The temporal conditional dependency expressed by $p(z_i \mid z_{<i})$ is commonly referred to as **temporal causality**, i.e., the causal dependency structure along the time axis.
>
> **The Impact of lacking Temporal Causality**
>
> Prior temporal-decomposition approaches (I2V-AR and Temporal Co-denoising) do not explicitly follow the temporal-causality modeling paradigm—namely, they do not attempt to model $p(z_i \mid z_{<i})$ . Instead, they rely on approximate or indirect strategies to capture the relationships between segments.
>
> - I2V-AR: They assume a first-order Markov property between segments and approximate the full temporal-causality factorization by conditioning only on the last frame of the previous segment. Formally, we have: $p(z_{1:N}) = \prod_{i=1}^{N} p(z_i \mid z_{<i})
> \approx \prod_{i=1}^{N} p(z_i \mid z_{i-1})
> \approx \prod_{i=1}^{N} p\bigl(z_i\mid \text{frame}(z_{i-1})\bigr).$
>
>     This simplification breaks temporal causality, as consecutive segments become only **visually** consistent while remaining **blind to the dynamics** of earlier segments. When a frame admits multiple plausible motion directions (e.g., a hand resting on a laptop could correspond to either opening or closing it), the model has no access to past dynamics and thus may infer inconsistent actions across segments, leading to **action-direction inconsistency**.
>
> - Temporal Co-denoising: It models long videos based on **temporal correlations** rather than causality. Formally, it approximates long video distribution via:
>
>     $p(z_{1:N}) \approx \prod_{i=1}^{N} p(z_i \mid z_i \cap z_{i-1})p(z_{i-1} \mid z_i \cap z_{i-1})$
>
>     **Each segment is generated independently,** relying only on the overlapping portion to maintain temporal correlation between adjacent segments. As a result, the two **independently generated segments may contain conflicting action directions internally**, which can lead to **action-direction conflicts** within the overlapping region.
>
>
> **Our InstructVC framework to model Temporal Causality**
>
> The chain-rule–based causal decomposition paradigm has achieved great success in sequence modeling. However, video sequences are extremely long (e.g., even a 5-second 480p video contains nearly 100k tokens after a 4×8×8 spatiotemporal VAE compression), and current long-sequence modeling capacity and computational budgets are insufficient to directly model temporal causality $p(z_i \mid z_{<i})$  at the video-token level. **Effective approximations of temporal causality modeling are needed.**
>
> Therefore, InstructVC leverages the complementary properties of the text and video modalities: it performs **global causal modeling in the text domain (Temporal Action Binding)** to ensure overall semantic consistency, while applying **local causal modeling in the video domain (Causal Video Continuation)** to maintain temporal continuity between adjacent segments. **This combination allows us to effectively approximate temporal causality despite current limitations in long-sequence video modeling.**
>
> **W2: Videos for figure 1**
>
> Thank you for the suggestion. We have included the corresponding videos in the appendix to better illustrate the motivation.

---

> ### Author Response · Authors · 2025-11-21
> **Thank you for careful review (2/2)**
>
> **W1: Improve the clarity of the third paragraph of the intro.**
>
> We apologize for the confusion and have improved the introduction in the revised version. In the third paragraph of the Introduction, our intention was to provide an overview of prior work on multi-action long-video generation and to highlight their limitations, which naturally motivates our approach. The temporal-expanding methods were mentioned mainly for completeness, while the focus is on the **temporal decomposition** family of approaches. **Both temporal co-denoising and I2V-AR, as well as our proposed SteinsGate, fall under this category.**  Compared with I2V-AR and temporal co-denoising, our method introduces a stronger notion of **temporal causality, which is the more principled factorization** $p(\text{video}) = p(\text{segment}_1) p(\text{segment}_2 \mid \text{segment}_1)$. This allows us to align and merge auto-regressively generated video segments, producing longer videos with improved temporal coherence.
>
> Furthermore, due to current compute and long-sequence modeling limits, temporal causality in video can practically be modeled only between adjacent segments. However, a full temporal causal chain-rule decomposition should not be restricted to just two neighboring segments — it should account for the entire past context. **To address this, we propose performing causal modeling and global planning in a much more compact text modality**: we plan and map the temporal sequence of actions and their predicted durations onto per-segment textual conditions as ***temporal control.*** This text-level planning provides a global context that supplements the local, adjacent-segment causal modeling in the video domain, enabling coherent long-range behavior despite the video model’s limited capacity for long temporal chains.
>
> **W4: Improve the clarity of the fourth and fifth paragraph of the intro.**
>
> We apologize for the confusion, and we have improved the clarity of the corresponding section. In short, InstructVC is **the overall framework including two steps below.** Following it, we propose **SteinsGate as a training-free proof-of-concept based on pretrained models**, built around an MLLM and the new temporal guidance method, Video Path Integral. At the implementation level, we also introduce **three practical heuristics above to further improve efficiency.**
>
> As noted earlier, our motivation stems from the insufficient modeling of temporal causality $p(z_i \mid z_{<i})$  in prior work. To address this, we propose the InstructVC framework, which estimates temporal causality through **global text-level modeling** and **local video-level modeling**. This involves two key steps:
>
> - **Temporal Action Binding.** In the abstract and compact text modality, we perform global planning and causal reasoning: based on all past actions, we determine what action should occur in each segment and how long it should last, and then inject this information into the per-segment prompts. With such dense temporal control, the global causal structure is preserved through the conditioning signals.
> - **Causal Video Continuation.** Given the text-level causal plan, the video model can focus on generating the current segment while maintaining continuity with the previous one (video continuation), without needing to model long-range video-level dependencies directly.
>
> SteinsGate is a **training-free instantiation** of this framework, serving as a proof of concept. **It uses an MLLM to** **perform Temporal Action Binding**, and employs our novel Video Path Integral guidance technique to convert a pretrained TI2V model into a video continuation model at inference time (**to perform** **Causal Video Continuation**). **To further improve both the sampling speed and the generation quality of Video Path Integral** when built on pre-trained TI2V models, we introduce **three practical heuristics:** Guidance Interval, History-aligned Redistribution, and Path Convergence Guidance.
>
> **W5: Evaluation Setting**
>
> For multi-action long-video evaluation, we follow MinT, whose setting closely matches our task. Multi-action long-video generation is indeed highly challenging, we start with a simple multi-action, no-scene-change setting to evaluate **whether** the MLLM can perform global temporal planning and **whether** a pre-trained TI2V model can be converted into a video continuation model at inference.
>
> Even under this simple setting, existing models—including those trained at large scale—still struggle to produce satisfactory results. As shown in Table 1, SteinsGate achieves a substantial improvement over training-free baselines such as DiTCtrl and FIFO Diffusion, and performs on par with large-scale models explicitly trained for video continuation, such as SkyReel-V2 and MAGI-1, demonstrating its strong relative performance. Furthermore, as foundation models and Video LLMs scale, our **plug-and-play framework** can naturally improve without expensive large-scaling re-training.

---

### Official Review · Reviewer_nDSM · 2025-11-01

**Soundness:** 4
**Presentation:** 3
**Contribution:** 3
**Rating:** 6
**Confidence:** 5

**Summary:**

This paper proposes a new framework, Instruct-Video-Continuation (InstructVC), to address the challenge of generating coherent, multi-action long videos. The framework contains two stages: Temporal Action Binding, which decomposes a user prompt into a sequence of segment plans with durations via LLM, and Causal Video Continuation, which autoregressively generates the video according to that plan. The authors introduce SteinsGate, a training-free, plug-and-play method of this framework that introduces a novel temporal guidance technique called Video Path Integral (VPI). VPI enforces causality by integrating multiple I2V video paths from different history frames.

**Strengths:**

1. The method is training-free and plug-and-play. It does not need the training resources, thus it could be integrated easily into methods.
2. The novelty of the VPI technique is quite sound and reasonable. The writing is also good.
3. The method is somewhat time-consuming when inference, while the authors recognise this and propose a speeding strategy of Guidance Interval GI. From the ablation study, this GI method achieves comparable results before and after speedup.

**Weaknesses:**

Overall, the paper is logically rigorous, addresses an important problem, and proposes a novel method. I do not see major flaws. However, there are a few minor points that, if answered by the authors, I would consider revising my score after rebuttal:
1. While VPI is a training-free method, it appears to introduce significant inference-time overhead. It will be better to quantify the increase in computational cost. Specifically, how many times more computationally expensive is the VPI method (which samples and integrates $K$ frames) compared to the baseline of generating from only the last frame?
2. The Path Convergence Guidance (PCG) method introduces a new hyperparameter, $w_1$. The paper seems to suggest setting this parameter to 5.0, but lacks a detailed explanation. While the concept may be intuitive to some, why must $w_1$ be a value greater than 1 (implying "over-correction")? Why can it not be less than 1 (implying a smoothing correction, similar to an EMA) or exactly 1 (implying a full replacement of the weak model's path)? The paper would be strengthened by a qualitative analysis, or ideally, an ablation study or visualization that justifies this specific choice.
3. The paper provides very little detail on how the "segment history" is selected. The Appendix (L657) only mentions that it is chosen based on a "fixed ratio" and that the frame count must satisfy a "$4N+1$" format. Why for both of these? Intuitively, the choice of history length seems highly sensitive:
- A history that is too short may prevent VPI from capturing sufficient information to continue the motion naturally.
- A history that is too long might make it extremely difficult to align a video generated from a very early frame with the distant future, potentially conflicting with the HR mechanism.
4. The method of introducing different prompts for different temporal segments (with or without an MLLM) is not entirely uncommon. The paper may be missing citations to some related works that have explored similar multi-prompt temporal decomposition strategies:

[1] Yan, Xin, et al. "Long video diffusion generation with segmented cross-attention and content-rich video data curation." Proceedings of the Computer Vision and Pattern Recognition Conference. 2025.

[2] Bansal, Hritik, et al. "Talc: Time-aligned captions for multi-scene text-to-video generation." arXiv preprint arXiv:2405.04682 (2024).

[3] Oh, Gyeongrok, et al. "MEVG: Multi-event Video Generation with Text-to-Video Models." ArXiv, 2023, https://arxiv.org/abs/2312.04086.

[4] Villegas, Ruben, et al. "Phenaki: Variable Length Video Generation From Open Domain Textual Description." ArXiv, 2022, https://arxiv.org/abs/2210.02399

**Questions:**

A baseline could be directly using the last frame of the last segment, with a new prompt, to obtain the new segment. And concat all segments together. Is this setting the same as the w/o VPI results?

---

> ### Author Response · Authors · 2025-11-21
> **Thank you for warm words and recognition !**
>
> Thanks for your warm words and recognition! We have updated the appendix according to your suggestions (Marked by blue). Sections with significant changes are marked by coloring their titles blue, while the content remains uncolored for clarity. We would like to address your concerns as follows:
>
> **W1: The time overhead**
>
> Sorry for the confusion.  We conducted the time evaluations on the same prompt set (average duration 15s, 225 frames) using identical pre-trained models and the same codebase. Under this controlled setup, I2V-AR takes approximately 30 minutes to generate a 15-second video, while SteinsGate requires around 38 minutes, **an acceptable ~25% inference time overhead in practice**.
>
> With guidance-interval and Monte Carlo estimation that randomly samples ~2 historical frames in single-step, **the overhead of integrating multiple I2V paths is amortized into each step**. Consequently, the practical cost of mixing different I2V paths in Video Path Integral is significantly reduced, achieving an effective balance between efficiency and performance.
>
> **W2: The ablation on the weight of Path Convergence Guidance**
>
> Sorry for the confusion, and we have updated the paper to include additional analysis and ablations on Path Convergence Guidance. Similar to Classifier-Free Guidance and AutoGuidance, weak-to-strong guidance methods generally require an extrapolative formulation, i.e., using a guidance weight larger than 1. The key intuition is that interpolation between the weak and strong directions often leads to worse results than using the strong direction alone, whereas **extrapolation shifts further along the ‘weak-to-strong’ direction $v_{\text{strong}} - v_{\text{weak}}$** (as shown in Fig4c), **typically yielding outputs better than using the strong direction alone.**
>
> Theoretically, this can be understood from the perspective of distribution shift: the guided direction corresponds to a modified distribution $P(x) P(\text{good} \mid x)^w$. **Only when the exponent w > 1 does the distribution shift sufficiently toward the desired region, enabling effective guidance.**
>
> **W3: On the selection of “segment history”**
>
> Sorry for any confusion.  First, we would like to clarify that the auto-selection of optimal video contexts for long video generation is itself an important research problem and it will rely on the understanding of previous video contexts and video-to-be-generated. And our work focuses on providing a training-free proof-of-concept for the InstructVC formulation to multi-action long video generation, **specifically on how to convert a pre-trained video generation model into a video continuation model.**
>
> Therefore, we adopt a simple, empirical strategy for selecting the historical segment: we set the number of historical frames to approximately **20% of the length of the upcoming segment** $(N_{\text{history}} \approx 0.2N_{\text{current}})$. **This choice aims to balance historical conditioning with text-driven control.** Since the Wan2.1 model can process at most ~81 frames per inference, using too many historical frames would reduce the available capacity for generating the new segment, thereby harming text adherence. Through empirical evaluation, we find that allocating ~**20% history and ~80% newly generated frames** provides a good trade-off. Because the generation length per step varies (typically 49–81 frames), using a relative ratio is more appropriate. This results in using roughly 13–25 historical frames—sufficient to preserve temporal dynamics while avoiding excessive constraints on the upcoming motion.
>
> Finally, the requirement that the video length follow the **4N+1** format is imposed by the underlying pre-trained video generation model. The current Video VAE (WanVAE in our work) encodes the first frame independently and then compresses every subsequent **four frames into one latent frame**, which necessitates that the generated video length be of the form 4N+1. To remain aligned with the base model, **we follow the same constraint: the historical segment is constructed to satisfy the 4N+1 format, and the newly generated segment follows the 4N format, so that the combined sequence (history + new frames) also conforms to the required 4N+1 structure.**
>
> **W4: Citation on more related works**
>
> Thank you for the suggestion! We have added citations and discussion of these works in the revised version. We would also like to clarify that the core idea of **Temporal Action Binding** is to decompose different actions into separate prompts and explicitly predict their temporal durations. This objective is fundamentally different from previous multi-prompt methods (e.g. for multi-scene generation), both in motivation and in function.
>
> **Q1: Baseline using the last frame of the last segment, with a new prompt, to obtain the new segment.**
>
> Sorry for the confusion! Yes, the baseline you mentioned corresponds exactly to the **w/o VPI** setting used in our experiments.

---

### Author Response · Authors · 2025-12-01
**A brief summary of the key pros and cons for the convenience of (senior) area chairs**

Dear (Senior) ACs,

We extend our sincere gratitude for the time and effort you have dedicated to reviewing our manuscript. In light of the recent rebuttal rollback due to information leakage, we have prepared a concise summary of our paper, the reviewers' comments, and our responses. We hope this consolidation helps you quickly understand and evaluate our submission.

**Our contribution is a new framework for multi-action long-video generation, InstructVC,** along with **a proof-of-concept implementation, SteinsGate.** SteinsGate first uses an MLLM as the video “actor” to decide the causal order, content and duration of each action (Temporal Action Binding in InstructVC). It then includes a plug-and-play temporal guidance method, Video Path Integral, which turns pre-trained text/image-to-video models into video-continuation models, enabling action-by-action, causal long video rendering (Causal Video Continuation in InstructVC).

We appreciate the reviewers' recognition of our work and we would like to summarize the key pros and cons noted by the reviewers, along with our responses, for your convenience:

> **Key pros noted by the reviewers:**
>

**S1:** Targeting an Important problem with a well-motivated new framework (Reviewer 8cn5, inkc).

**S2:** The plug-and-play nature of Video Path Integral makes it easy to combine with different base models (Reviewer nDSM, inkc).

**S3:** The novelty and effectiveness of Video Path Integral (Reviewer nDSM, inkc).

**S4:** The integration of MLLM for temporally grounding and temporal control (Reviewer inkc).

**S5:** The clear writings, illustrations and experiments (Reviewer nDSM, 8cn5, inkc).

**S6:** Practical techniques to further improve Video Path Integral (Reviewer nDSM, inkc).

> **Key cons noted by the reviewers and our response:**
>

**W1:** More Inference-time Analysis (Reviewer nDSM, inkc)

We have added inference-time comparison in the updated manuscript.

**W2:** Further Implementation Details (Reviewer nDSM, inkc)

We have added further implementation details, including algorithm details and pseudo codes, details of benchmark constructions and hyper-parameter choices, etc.

**W3:** Better presentations of paragraphs 3–5 of the Introduction (Reviewer 8cn5)

We have provided a clearer and more concise introduction in the revised paper and more video demos to demonstrate our motivations and performance.

**W4:** More ablations on hyper-parameters (Reviewer nDSM, inkc)

We have included a detailed ablations for key hyper-parameters in the revised appendix.

**W5:** More theoretical analysis of video path integral (Reviewer inkc).

We have provided a comprehensive theoretical analysis for video path integral during rebuttal.

We sincerely appreciate the reviewers’ recognition of our efforts and their constructive suggestions, which have helped strengthen the manuscript. We are especially grateful for your time and dedication, particularly given the recent surge in workload. We deeply respect the effort it takes to maintain such careful attention to detail for every submission under these challenging circumstances.

Warm regards,

Authors

---

### Meta-Review · Area_Chair_iHkh · 2026-01-05

**Summary:**

The manuscript presents a framework, InstructVC, to generate long multi-action videos. InstructVC consists of two stages: Temporal Action Binding, which decomposes a user prompt into a sequence of segment plans with durations via LLM, and Causal Video Continuation, which generates the video in an autoregressive way. SteinsGate is practical implementaiton of InstructVC. It first uses an MLLM as the video “actor” to decide the causal order. It introduces a temporal guidance technique, Video Path Integral (VPI), that enforces causality by integrating multiple I2V video paths from different history frames. The method is training-free and plug-and-play and shows good results.

**Reviewer Concerns:**

The review scores were quite positive from the start, with some concerns regarding clarity of writing, formal definitions, example videos, and ablations.
In particular, questions concern:
1. Inference time overhead
2. Ablations on newly introduced hyperparameters
3. Details on the "segment history"
and further implementation details

The rebuttal addresses most of these questions, provides necessary formalizations, implementaiton details, ablations and the necessary clarifications on the technical paragraphs of the paper.

**Reviewer Scores:**

This is hard to say. My expectation is that  reviewer inkc would stick to the score of 8 originally provided while both other reviewers might have given a score of 6. The rebuttal addresses all relevant questions and there were no major flaws outlined in the reviews that would be prohibitive.

---

### Decision · Program_Chairs · 2026-01-26

Accept (Poster)